# Group Additive Structure Identification for Kernel Nonparametric Regression

**Pan Chao**
Department of Statistics
Purdue University
West Lafayette, IN 47906
panchao25@gmail.com

**Michael Zhu**
Department of Statistics, Purdue University
West Lafayette, IN 47906
Center for Statistical Science
Department of Industrial Engineering
Tsinghua University, Beijing, China
yuzhu@purdue.edu

## Abstract

The additive model is one of the most popularly used models for high dimensional nonparametric regression analysis. However, its main drawback is that it neglects possible interactions between predictor variables. In this paper, we reexamine the group additive model proposed in the literature, and rigorously define the intrinsic group additive structure for the relationship between the response variable $Y$ and the predictor vector $\boldsymbol{X}$, and further develop an effective structure-penalized kernel method for simultaneous identification of the intrinsic group additive structure and nonparametric function estimation. The method utilizes a novel complexity measure we derive for group additive structures. We show that the proposed method is consistent in identifying the intrinsic group additive structure. Simulation study and real data applications demonstrate the effectiveness of the proposed method as a general tool for high dimensional nonparametric regression.

## 1 Introduction

Regression analysis is popularly used to study the relationship between a response variable $Y$ and a vector of predictor variables $\boldsymbol{X}$. Linear and logistic regression analysis are arguably two most popularly used regression tools in practice, and both postulate explicit parametric models on $f(\boldsymbol{X}) = \mathbb{E}[Y|\boldsymbol{X}]$ as a function of $\boldsymbol{X}$. When no parametric models can be imposed, nonparametric regression analysis can instead be performed. On one hand, nonparametric regression analysis is flexible and not susceptible to model mis-specification, whereas on the other hand, it suffers from a number of well-known drawbacks especially in high dimensional settings. Firstly, the asymptotic error rate of nonparametric regression deteriorates quickly as the dimension of $\boldsymbol{X}$ increases. [16] shows that with some regularity conditions, the optimal asymptotic error rate for estimating a $d$-times differentiable function is $\mathcal{O}\left(n^{-d/(2d+p)}\right)$, where $p$ is the dimensionality of $\boldsymbol{X}$. Secondly, the resulting fitted nonparametric function is often complicated and difficult to interpret.

To overcome the drawbacks of high dimensional nonparametric regression, one popularly used approach is to impose the additive structure [5] on $f(\boldsymbol{X})$, that is to assume that $f(\boldsymbol{X}) = f_1(X_1) + \cdots + f_p(X_p)$ where $f_1, \ldots, f_p$ are $p$ unspecified univariate functions. Thanks to the additive structure, the nonparametric estimation of $f$ or equivalently the individual $f_i$'s for $1 \leq i \leq p$ becomes efficient and does not suffer from the curse of dimensionality. Furthermore, the interpretability of the resulting model has also been much improved.

The key drawback of the additive model is that it does not assume interactions between the predictor variables. To address this limitation, functional ANOVA models were proposed to accommodate higher order interactions, see [4] and [13]. For example, by neglecting interactions of

order higher than 2, the functional ANOVA model can be written as $f(\boldsymbol{X}) = \sum_{i=1}^{p} f_i(X_i) + \sum_{1 \le i,j \le p} f_{ij}(X_i, X_j)$, with some marginal constraints. Another higher order interaction model, $f(\boldsymbol{X}) = \sum_{d=1}^{D} \sum_{1 \le i_1, \ldots, i_d \le p} f_j(X_{i_1}, \ldots, X_{i_d})$, is proposed by [6]. This model considers all interactions of order up to $D$, which is estimated by Kernel Ridge Regression (KRR) [10] with the elementary symmetric polynomial (ESP) kernel. Both of the two models assume the existence of possible interactions between any two or more predictor variables. This can lead to a serious problem, that is, the number of nonparametric functions that need to be estimated quickly increases as the number of predictor variables increases.

There exists another direction to generalize the additive model. When proposing the Optimal Kernel Group Transformation (OKGT) method for nonparametric regression, [11] considers the additive structure of predictor variables in groups instead of individual predictor variables. Let $G := \{\boldsymbol{u}_j\}_{j=1}^{d}$ be a index partition of the predictor variables, that is, $\boldsymbol{u}_j \cap \boldsymbol{u}_k = \emptyset$ if $j \ne k$ and $\cup_{j=1}^{d} \boldsymbol{u}_j = \{1, \ldots, p\}$. Let $\boldsymbol{X}_{\boldsymbol{u}_j} = \{X_k; k \in \boldsymbol{u}_j\}$ for $j = 1, \ldots, d$. Then $\{X_1, \ldots, X_d\} = \boldsymbol{X}_{\boldsymbol{u}_1} \cup \cdots \cup \boldsymbol{X}_{\boldsymbol{u}_d}$. For any function $f(\boldsymbol{X})$, if there exists an index partition $G = \{\boldsymbol{u}_1, \ldots, \boldsymbol{u}_d\}$ such that

$$f(\boldsymbol{X}) = f_{\boldsymbol{u}_1}(\boldsymbol{X}_{\boldsymbol{u}_1}) + \ldots + f_{\boldsymbol{u}_d}(\boldsymbol{X}_{\boldsymbol{u}_d}), \tag{1}$$

where $f_{\boldsymbol{u}_1}(\boldsymbol{X}_{\boldsymbol{u}_1}), \ldots, f_{\boldsymbol{u}_d}(\boldsymbol{X}_{\boldsymbol{u}_d})$ are $d$ unspecified nonparametric functions, then it is said that $f(\boldsymbol{X})$ admits the *group additive structure* $G$. We also refer to (1) as a group additive model for $f(\boldsymbol{X})$. It is clear that the usual additive model is a special case with $G = \{(1), \ldots, (p)\}$.

Suppose $X_{j_1}$ and $X_{j_2}$ are two predictor variables. Intuitively, if $X_{j_1}$ and $X_{j_2}$ interact to each other, then they must appear in the same group in an reasonable group additive structure of $f(\boldsymbol{X})$. On the other hand, if $X_{j_1}$ and $X_{j_2}$ belong to two different groups, then they do not interact with each other. Therefore, in terms of accommodating interactions, the group additive model can be considered lying in the middle between the original additive model and the functional ANOVA or higher order interaction models. When the group sizes are small, for example all are less than or equal to 3, the group additive model can maintain the estimation efficiency and interpretability of the original additive model while avoiding the problem of a high order model discussed earlier.

However, in [11], there are two important issues not addressed. First, the group additive structure may not be unique, which will lead to the nonidentifiability problem for the group additive model. (See discussion in Section 2.1). Second, [11] has not proposed a systematic approach to identify the group additive structure. In this paper, we intend to resolve these two issues. To address the first issue, we rigorously define the *intrinsic group additive structure* for any square integrable function, which in some sense is the minimal group additive structure among all correct group additive structures.

To address the second issue, we propose a general approach to simultaneously identifying the intrinsic group additive structure and estimating the nonparametric functions using kernel methods and Reproducing Kernel Hilbert Spaces (RKHSs). For a given group additive structure $G = \{\boldsymbol{u}_1, \ldots, \boldsymbol{u}_d\}$, we first define the corresponding direct sum RKHS as $\mathcal{H}_G = \mathcal{H}_{\boldsymbol{u}_1} \oplus \cdots \oplus \mathcal{H}_{\boldsymbol{u}_d}$ where $\mathcal{H}_{\boldsymbol{u}_i}$ is the usual RKHS for the variables in $\boldsymbol{u}_j$ only for $j = 1, \ldots, d$. Based on the results on the capacity measure of RKHSs in the literature, we derive a tractable capacity measure of the direct sum RKHS $\mathcal{H}_G$ which is further used as the complexity measure of $G$. Then, the identification of the intrinsic group additive structure and the estimation of the nonparametric functions can be performed through the following minimization problem

$$\hat{f}, \hat{G} = \operatorname*{arg\,min}_{f \in \mathcal{H}_{G}, G} \frac{1}{n} \sum_{i=1}^{n} (y_i - f(\boldsymbol{x}_i))^2 + \lambda \|f\|_{\mathcal{H}_G}^2 + \mu \boldsymbol{\mathcal{C}}(G).$$

We show that when the novel complexity measure of group additive structure $\boldsymbol{\mathcal{C}}(G)$ is used, the minimizer $\hat{G}$ is consistent for the intrinsic group additive structure. We further develop two algorithms, one uses exhaustive search and the other employs a stepwise approach, for identifying true additive group structures under the small $p$ and large $p$ scenarios. Extensive simulation study and real data applications show that our proposed method can successfully recover the true additive group structures in a variety of model settings.

There exists a connection between our proposed group additive model and graphical models ([2], [7]). This is especially true when a sparse block structure is imposed [9]. However, a key difference exists. Let's consider the following example. $Y = \sin(X_1 + X_2^2 + X_3) + \cos(X_4 + X_5 + X_6^2) + \epsilon$. A graphical model typically considers the conditional dependence (CD) structure among all of the

variables including $X_1, \ldots, X_6$ and $Y$, which is more complex than the group additive (GA) structure $\{(X_1, X_2, X_3), (X_4, X_5, X_6)\}$. The CD structure, once known, can be further examined to infer the GA structure. In this paper, we however proposed methods that directly target the GA structure instead of the more complex CD structure.

The rest of the paper is organized as follows. In Section 2, we rigorously formulate the problem of Group Additive Structure Identification (GASI) for nonparametric regression and propose the structural penalty method to solve the problem. In Section 3, we prove the selection consistency for the method. We report the experimental results based on simulation studies and real data application in Section 4 and 5. Section 6 concludes this paper with discussion.

## 2 Method

### 2.1 Group Additive Structures

In the Introduction, we discussed that the group additive structure for $f(\boldsymbol{X})$ may not be unique. Here is an example. Consider the model $Y = 2 + 3X_1 + 1/(1 + X_2^2 + X_3^2) + \arcsin((X_4 + X_5)/2) + \epsilon$, where $\epsilon$ is the 0 mean error independent of $\boldsymbol{X}$. According to our definition, this model admits the group additive structure $G_0 = \{(1), (2, 3), (4, 5)\}$. Let $G_1 = \{(1, 2, 3), (4, 5)\}$ and $G_2 = \{(1, 4, 5), (2, 3)\}$. The model can also be said to admit $G_1$ and $G_2$. However, there exists a major difference between $G_0$, $G_1$ and $G_2$. While the groups in $G_0$ cannot be further divided into subgroups, both $G_1$ and $G_2$ contain groups that can be further split. We define the following partial order between group structures to characterize the difference and their relationship.

**Definition 1.** *Let $G$ and $G'$ be two group additive structures. If for every group $\boldsymbol{u} \in G$ there is a group $\boldsymbol{v} \in G'$ such that $\boldsymbol{u} \subseteq \boldsymbol{v}$, then $G$ is called a **sub group additive structure** of $G'$. This relation is denoted as $G \leq G'$. Equivalently, $G'$ is a **super group additive structure** of $G$, denoted as $G' \geq G$.*

In the previous example, $G_0$ is a sub group additive structure of both $G_1$ and $G_2$. However, the order between $G_1$ and $G_2$ is not defined. Let $\mathcal{X} := [0, 1]^p$ be the p-dimensional unit cube for all the predictor variables $\boldsymbol{X}$ with distribution $P_{\boldsymbol{X}}$. For a group of predictor variables $\boldsymbol{u}$, we define the space of square integrable functions as $L_{\boldsymbol{u}}^2(\mathcal{X}) := \{g \in L_{P_{\boldsymbol{X}}}^2(\mathcal{X}) \mid g(\boldsymbol{X}) = f_{\boldsymbol{u}}(\boldsymbol{X}_{\boldsymbol{u}})\}$, that is $L_{\boldsymbol{u}}^2$ contains the functions that only depend on the variables in group $\boldsymbol{u}$. Then the group additive model $f(\boldsymbol{X}) = \sum_{j=1}^d f_{\boldsymbol{u}_j}(\boldsymbol{X}_{\boldsymbol{u}_j})$ is a member of the direct sum function space defined as $L_G^2(\mathcal{X}) := \oplus_{\boldsymbol{u} \in G} L_{\boldsymbol{u}}^2(\mathcal{X})$. Let $|\boldsymbol{u}|$ be the cardinality of the group $\boldsymbol{u}$. If $\boldsymbol{u}$ is the only group in a group additive structure and $|\boldsymbol{u}| = p$, then $L_{\boldsymbol{u}}^2 = L_G^2$ and $f$ is a fully non-parametric function.

The following proposition shows that the order of two different group additive structures is preserved by their corresponding square integrable function spaces.

**Proposition 1.** *Let $G_1$ and $G_2$ be two group additive structures. If $G_1 \leq G_2$, then $L_{G_1}^2 \subseteq L_{G_2}^2$. Furthermore, if $X_1, \ldots, X_p$ are independent and $G_1 \neq G_2$, then $L_{G_1}^2 \subset L_{G_2}^2$.*

**Definition 2.** *Let $f(\boldsymbol{X})$ be an square integrable function. For a group additive structure $G$, if there is a function $f_G \in L_G^2$ such that $f_G = f$, then $G$ is called an **amiable group additive structure** for $f$.*

In the example discussed in the beginning of the subsection, $G_0$, $G_1$ and $G_2$ are all amiable group structures. So amiable group structures may not be unique.

**Proposition 2.** *Suppose $G$ is an amiable group additive structure for $f$. If there is a second group additive structure $G'$ such that $G \leq G'$, then $G'$ is also amiable for $f$.*

We denote the collection of all amiable group structures for $f(\boldsymbol{X})$ as $\mathcal{G}^a$, which is partially ordered and complete. Therefore, there exists a minimal group additive structure in $\mathcal{G}^a$, which is the most concise group additive structure for the target function. We state this result as a theorem.

**Theorem 1.** *Let $\mathcal{G}^a$ be the set of amiable group additive structures for $f$. There is a unique minimal group additive structure $G^* \in \mathcal{G}^a$ such that $G^* \leq G$ for all $G \in \mathcal{G}^a$, where the order is given by Definition 1. $G^*$ is called the **intrinsic group additive structure** for $f$.*

For statistical modeling, $G^*$ achieves the greatest dimension reduction for the relationship between $Y$ and $\boldsymbol{X}$. It induces the smallest function space which includes the model. In general, the intrinsic group structure can help much mitigate the curse of dimensionality while improving both efficiency and interpretability of high dimensional nonparametric regression.

## 2.2 Kernel Method with Known Intrinsic Group Additive Structure

Suppose the intrinsic group additive structure for $f(\boldsymbol{X}) = \mathbb{E}[Y|\boldsymbol{X}]$ is known to be $G^* = \{\boldsymbol{u}_j\}_{j=1}^d$, that is, $f(\boldsymbol{X}) = f_{\boldsymbol{u}_1}(\boldsymbol{X}_{\boldsymbol{u}_1}) + \cdots + f_{\boldsymbol{u}_d}(\boldsymbol{X}_{\boldsymbol{u}_d})$. Then, we will use the kernel method to estimate the functions $f_{\boldsymbol{u}_1}, f_{\boldsymbol{u}_2}, \ldots, f_{\boldsymbol{u}_d}$. Let $(K_{\boldsymbol{u}_j}, \mathcal{H}_{\boldsymbol{u}_j})$ be the kernel and its corresponding RKHS for the $j$-th group $\boldsymbol{u}_j$. Then using kernel methods is to solve

$$\hat{f}_{\lambda, G^*} = \underset{f_{G^*} \in \mathcal{H}_{G^*}}{\arg\min} \left\{ \frac{1}{n} \sum_{i=1}^n (y_i - f_{G^*}(\boldsymbol{x}_i))^2 + \lambda \|f_{G^*}\|_{\mathcal{H}_{G^*}}^2 \right\}, \tag{2}$$

where $\mathcal{H}_{G^*} := \{f = \sum_{j=1}^d f_{\boldsymbol{u}_j} \mid f_{\boldsymbol{u}_j} \in \mathcal{H}_{\boldsymbol{u}_j}\}$. The subscripts of $\hat{f}$ are used to explicitly indicate its dependence on the group additive structure $G^*$ and tuning parameter $\lambda$.

In general, an RKHS is usually smaller than the $L^2$ space defined on the same input domain. So, it is not always true that $\hat{f}_{\lambda, G^*}$ achieves $f$. However, one can choose to use universal kernels $K_{\boldsymbol{u}_j}$ so that their corresponding RKHSs are dense in the $L^2$ spaces (see [3], [15]). Using universal kernels allows $\hat{f}_{\lambda, G^*}$ to not only achieve unbiasedness but also recover the group additive structure of $f(\boldsymbol{X})$. This is the fundamental reason for the consistency property of our proposed method to identify the intrinsic group additive structure. Two examples of universal kernel are Gaussian and Laplace.

## 2.3 Identification of Unknown Intrinsic Group Additive Structure

### 2.3.1 Penalization on Group Additive Structures

The success of the kernel method hinges on knowing the intrinsic group additive structure $G^*$. In practice, however, $G^*$ is seldom known, and it may be of primary interest to identify $G^*$ while estimating a group additive model. Recall that in Subsection 2.1, we have shown that $G^*$ exists and is unique. The other group additive structures belong to two categories, amiable and non-amiable.

Let's consider an arbitrary non-amiable group additive structure $G \in \mathcal{G} \setminus \mathcal{G}^a$ first. If $G$ is used in the place of $G^*$ in (2), the solution $\hat{f}_{\lambda, G}$, as an estimator of $f$, will have a systematic bias because the $L^2$ distance between any function $f_G \in \mathcal{H}_G$ and the true function $f$ will be bounded below. In other words, using a non-amiable structure will result in poor fitting of the model.

Next we consider an arbitrary amiable group additive structure $G \in \mathcal{G}^a$ to be used in (2). Recall that because $G$ is amiable, we have $f_{G^*} = f_G$ almost surely (in population) for the true function $f(\boldsymbol{X})$. The bias of the resulting fitted function $\hat{f}_{\lambda, G}$ will vanish as the sample size increases. Although their asymptotic rates are in general different, under fixed sample size $n$, simply using goodness of fit will not be able to distinguish $G$ from $G^*$. The key difference between $G^*$ and $G$ is their structural complexities, that is, $G^*$ is the smallest among all amiable structures (i.e. $G^* \leq G, \forall G \in \mathcal{G}^a$). Suppose a proper complexity measure of a group additive structure $G$ can be defined (to be addressed in the next section) and is denoted as $\mathcal{C}(G)$. We can then incorporate $\mathcal{C}(G)$ into (2) as an additional penalty term and change the kernel method to the following structure-penalized kernel method.

$$\hat{f}_{\lambda, \mu}, \hat{G} = \underset{f_G \in \mathcal{H}_G, G}{\arg\min} \left\{ \frac{1}{n} \sum_{i=1}^n (y_i - f_G(\boldsymbol{x}_i))^2 + \lambda \|f_G\|_{\mathcal{H}_G}^2 + \mu \mathcal{C}(G) \right\}. \tag{3}$$

It is clear that the only difference between (2) and (3) is the term $\mu \mathcal{C}(G)$. As discussed above, the intrinsic group additive structure $G^*$ can achieve the goodness of fit represented by the first two terms in (3) and the penalty on the structural complexity represented by the last term. Therefore, by properly choosing the tuning parameters, we expect that $\hat{G}$ is consistent in that the probability $\hat{G} = G^*$ increases to one as $n$ increases (see the Theory Section below). In the next section, we derive a tractable complexity measure for a group additive structure.

### 2.3.2 Complexity Measure of Group additive Structure

It is tempting to propose an intuitive complexity measure for a group additive structure $\mathcal{C}(\cdot)$ such that $\mathcal{C}(G_1) \leq \mathcal{C}(G_2)$ whenever $G_1 \leq G_2$. The intuition however breaks down or at least becomes less clear when the order between $G_1$ and $G_2$ cannot be defined. From Proposition 1, it is known that when $G_1 < G_2$, we have $L_{G_1}^2 \subset L_{G_2}^2$. It is not difficult to show that it is also true that when

$G_1 < G_2$, then $\mathcal{H}_{K,G_1} \subset \mathcal{H}_{K,G_2}$. This observation motivates us to define the structural complexity measure of $G$ through the capacity measure of its corresponding RKHS $\mathcal{H}_G$.

There exist a number of different capacity measures for RKHSs in the literature, including entropy [18], VC dimensions [17], Rademacher complexity [1], and covering numbers ([14], [18]). After investigating and comparing different measures, we use covering number to design a practically convenient complexity measure for group additive structures.

It is known that an RKHS $\mathcal{H}_K$ can be embedded in the continuous function space $\mathcal{C}(\mathcal{X})$ (see [12], [18]), with the inclusion mapping denoted as $I_K : \mathcal{H}_K \to \mathcal{C}(\mathcal{X})$. Let $\mathcal{H}_{K,r} = \{h : \|h\|_{H_k} \leq r,$ and $h \in \mathcal{H}_K\}$ be an $r$-ball in $\mathcal{H}_K$ and $\overline{I(\mathcal{H}_{K,r})}$ be the closure of $I(\mathcal{H}_{K,r}) \subseteq \mathcal{C}(\mathcal{X})$. One way to measure the capacity of $\mathcal{H}_K$ is through the covering number of $\overline{I(\mathcal{H}_{K,r})}$ in $\mathcal{C}(\mathcal{X})$, denoted as $\mathcal{N}(\epsilon, \overline{I(\mathcal{H}_{K,r})}, d_\infty)$, which is the smallest cardinalty of a subset $S$ of $\mathcal{C}(\mathcal{X})$ such that $\overline{I(\mathcal{H}_{K,r})} \subset \cup_{s \in S}\{t \in \mathcal{C}(\mathcal{X}) : d_\infty(t,s) \leq \epsilon\}$. Here $\epsilon$ is any small positive value and $d_\infty$ is the sup-norm.

The exact formula for $\mathcal{N}(\epsilon, \overline{I(\mathcal{H}_{K,r})}, d_\infty)$ is in general not available. Under certain conditions, various upper bounds have been obtained in the literature. One such upper bound is presented below.

When $K$ is a convolution kernel, i.e. $K(x,t) = k(x-t)$, and the Fourier transform of $k$ decays exponentially, then it is given in [18] that

$$\ln \mathcal{N}\left(\epsilon, \overline{I(\mathcal{H}_{K,r})}, d_\infty\right) \leq C_{k,p}\left(\ln \frac{r}{\epsilon}\right)^{p+1}, \tag{4}$$

where $C_{k,p}$ is a constant depending on the kernel function $k$ and input dimension $p$. In particular, when $K$ is a Gaussian kernel, [18] has obtained more elaborate upper bounds.

The upper bound in (4) depends on $r$ and $\epsilon$ through $\ln(r/\epsilon)$. When $\epsilon \to 0$ with $r$ being fixed (e,g. $r = 1$ when a unit ball is considered), $(\ln(r/\epsilon))^{p+1}$ dominates the upper bound. According to [8], the growth rate of $\mathcal{N}(\epsilon, I_K)$ or its logarithm can be viewed as a capacity measure of RKHS. So we use $(\ln(r/\epsilon))^{p+1}$ as the capacity measure, which can be reparameterized as $\alpha^{p+1}$ with $\alpha = \ln(r/\epsilon)$. Let $\mathcal{C}(\mathcal{H}_k)$ denote the capacity measure of $\mathcal{H}_k$, which is defined as $\mathcal{C}(\mathcal{H}_k) = (\ln(r/\epsilon))^{p+1} = \alpha(\epsilon)^{p+1}$. We know $\epsilon$ is the radius of a covering ball, which is the unit of measurement we use to quantify the capacity. The capacity of two RKHSs with different input dimensions are easier to be differentiated when $\epsilon$ is small. This gives an interpretation of $\alpha$.

We have defined a capacity measure for a general RKHS. In Problem (3), the model space $\mathcal{H}_G$ is a direct sum of a number of RKHSs. Let $G = \{\boldsymbol{u}_1, \ldots, \boldsymbol{u}_d\}$; let $\mathcal{H}_G, \mathcal{H}_{\boldsymbol{u}_1}, \ldots, \mathcal{H}_{\boldsymbol{u}_d}$ be the RKHSs corresponding to $G, \boldsymbol{u}_1, \ldots, \boldsymbol{u}_d$, respectively; let $I_G, I_{\boldsymbol{u}_1}, \ldots, I_{\boldsymbol{u}_d}$ be the inclusion mappings of $\mathcal{H}_G, \mathcal{H}_{\boldsymbol{u}_1}, \ldots, \mathcal{H}_{\boldsymbol{u}_d}$ into $\mathcal{C}(\mathcal{X})$. Then, we have the following proposition.

**Proposition 3.** *Let $G$ be a group additive structure and $\mathcal{H}_G$ be the induced direct sum RKHS defined in (3). Then, we have the following inequality relating the covering number of $\mathcal{H}_G$ and the covering numbers of $\mathcal{H}_{\boldsymbol{u}_j}$*

$$\ln \mathcal{N}(\epsilon, I_G, d_\infty) \leq \sum_{j=1}^{d} \ln \mathcal{N}\left(\frac{\epsilon}{|G|}, I_{\boldsymbol{u}_j}, d_\infty\right), \tag{5}$$

*where $|G|$ denotes the number of groups in $G$.*

By applying Proposition 3 and using the parameterized upper bound, we have $\ln \mathcal{N}(\epsilon, I_G, d_\infty) = \mathcal{O}\left(\sum_{\boldsymbol{u} \in G} \alpha(\epsilon)^{|\boldsymbol{u}|+1}\right)$. The rate can be used as the explicit expression of the complexity measure for group additive structures, that is $\mathcal{C}(G) = \sum_{j=1}^{d} \alpha(\epsilon)^{|\boldsymbol{u}_j|+1}$. Recall that there is another tuning parameter $\mu$ which controls the effect of the complexity of group structure on the penalized risk. By combining the common factor 1 in the exponent with $\mu$, we could further simplify the penalty's expression. Thus, we have the following explicit formulation for GASI

$$\hat{f}_{\lambda,\mu}, \hat{G} = \underset{f_G \in \mathcal{H}_G, G}{\arg\min} \left\{ \sum_{i=1}^{n} (y_i - f_G(\boldsymbol{x}_i))^2 + \lambda \|f_G\|_{\mathcal{H}_G}^2 + \mu \sum_{j=1}^{d} \alpha^{|\boldsymbol{u}_j|} \right\}. \tag{6}$$

## 2.4 Estimation

We assume that the value of $\lambda$ is given. In practice, $\lambda$ can be tuned separately. If the values of $\mu$ and $\alpha$ are also given, Problem (6) can be solved by following a two-step procedure.

First, when the group structure $G$ is known, $f_{\boldsymbol{u}}$ can be estimated by solving the following problem

$$\hat{\mathcal{R}}_G^\lambda = \min_{f_G \in \mathcal{H}_G} \left\{ \frac{1}{n} \sum_{i=1}^n (y_i - f_G(\boldsymbol{x}_i))^2 + \lambda \|f_G\|_{\mathcal{H}_G}^2 \right\}. \qquad (7)$$

Second, the optimal group structure is chosen to achieve both small fitting error and complexity, i.e.

$$\hat{G} = \arg\min_{G \in \mathcal{G}} \left\{ \hat{\mathcal{R}}_G^\lambda + \mu \sum_{j=1}^d \alpha^{|\boldsymbol{u}_j|} \right\}. \qquad (8)$$

The two-step procedure above is expected to identify the intrinsic group structure, that is, $\hat{G} = G^*$. Recall a group structure belongs to one of the three categories, intrinsic, amiable, or non-amiable structures. If $G$ is non-amiable, then $\hat{\mathcal{R}}_G^\lambda$ is expected to be large, because $G$ is a wrong structure which will result in a biased estimate. If $G$ is amiable, though $\hat{\mathcal{R}}_G^\lambda$ is expected to be small, the complexity penalty of $G$ is larger than that for $G^*$. As a consequence, only $G^*$ can simultaneously achieve a small $\hat{\mathcal{R}}_{G^*}^\lambda$ and a relatively small complexity. Therefore, when the sample size is large enough, we expect $\hat{G} = G^*$ with high probability. If the values of $\mu$ and $\alpha$ are not given, a separate validation set can be used to select tuning parameters.

The two-step estimation is summarized in Algorithm 1. When a model contains a large number of predictor variables, such exhaustive search suffers high computational cost. In order to apply GASI on a large model, we propose a backward stepwise algorithm which is illustrated in Algorithm 2.

| Algorithm 1:  Exhaustive Search w/ Validation |
| --- |
| 1: Split data into training ($\mathcal{T}$) and validation ($\mathcal{V}$) sets. |
| 2: **for** $(\mu, \alpha)$ in grid **do** |
| 3:     **for** $G \in \mathcal{G}$ **do** |
| 4:         $\hat{\mathcal{R}}_G, \hat{f}_G \leftarrow$ solve (7) using $G$; |
| 5:         Calculate the sum in (8), denoted by $\hat{\mathcal{R}}_G^{\text{pen},\mu,\alpha}$; |
| 6:     **end for** |
| 7:     $\hat{G}^{\mu,\alpha} \leftarrow \arg\min_{G \in \mathcal{G}} \hat{\mathcal{R}}_G^{\text{pen},\mu,\alpha}$; |
| 8:     $\hat{y}^{\mathcal{V}} \leftarrow \hat{f}_{\hat{G}^{\mu,\alpha}}(\boldsymbol{x}^{\mathcal{V}})$; |
| 9:     $e_{\hat{G}^{\mu,\alpha}}^2 \leftarrow \|y^{\mathcal{V}} - \hat{y}^{\mathcal{V}}\|^2$; |
| 10: **end for** |
| 11: $\mu^*, \alpha^* \leftarrow \arg\min_{\mu,\alpha} e_{\hat{G}^{\mu,\alpha}}^2$; |
| 12: $G^* \leftarrow \hat{G}^{\mu^*,\alpha^*}$; |

| Algorithm 2:  Basic Backward Stepwise |
| --- |
| 1: Start with the group structure $\{(1, \dots, p)\}$; |
| 2: Solve (6) and obtain its minimum value $\hat{\mathcal{R}}_G^{\text{pen}}$; |
| 3: **for** each predictor variable $j$ **do** |
| 4:     $G' \leftarrow$ either split $j$ as a new group or add to an existing group; |
| 5:     Solve (6) and obtain its minimum value $\hat{\mathcal{R}}_{G'}^{\text{pen}}$; |
| 6:     **if** $\hat{\mathcal{R}}_{G'}^{\text{pen}} < \hat{\mathcal{R}}_G^{\text{pen}}$ **then** |
| 7:         Keep $G'$ as the new group structure; |
| 8:     **end if** |
| 9: **end for** |
| 10: **return** $G'$; |

## 3  Theory

In this section, we prove that the estimated group additive structure $\hat{G}$ as a solution to (6) is consistent, that is the probability $P(\hat{G} = G^*)$ goes to 1 as the sample size $n$ goes to infinity. The proof and supporting lemmas are included in the supplementary material.

Let $\mathcal{R}(f_G) = \mathbb{E}[(Y - f(\boldsymbol{X}))^2]$ denote the population risk of a function $f \in \mathcal{H}_G$, and $\hat{\mathcal{R}}(f) = \frac{1}{n} \sum_{i=1}^n (y_i - f(\boldsymbol{x}_i))^2$ be the empirical risk. First, we show that for any amiable structure $G \in \mathcal{G}^a$, its minimized empirical risk $\hat{\mathcal{R}}(\hat{f}_G)$ converges in probability to the optimal population risk $\mathcal{R}(f_{G^*}^*)$ achieved by the intrinsic group additive structure. Here $\hat{f}_G$ denotes the minimizer of Problem (7) with the given $G$, and $f_{G^*}^*$ denotes the minimizer of the population risk when the intrinsic group structure is used. The result is given below as Proposition 4.

**Proposition 4.** *Let $G^*$ be the intrinsic group additive structure, $G \in \mathcal{G}^a$ a given amiable group structure, and $\mathcal{H}_{G^*}$ and $\mathcal{H}_G$ the respective direct sum RKHSs. If $\hat{f}_G^\lambda \in \mathcal{H}_G$ is the optimal solution of*

| ID | Model | Intrinsic Group Structure |
|---|---|---|
| M1 | $y = 2x_1 + x_2^2 + x_3^3 + \sin(\pi x_4) + \log(x_5 + 5) + |x_6| + \epsilon$ | $\{(1),(2),(3),(4),(5),(6)\}$ |
| M2 | $y = \frac{1}{1+x_1^2} + \arcsin\left(\frac{x_2+x_3}{2}\right) + \arctan\left((x_4 + x_5 + x_6)^3\right) + \epsilon$ | $\{(1),(2,3),(4,5,6)\}$ |
| M3 | $y = \arcsin\left(\frac{x_1+x_3}{2}\right) + \frac{1}{1+x_2^2} + \arctan\left((x_4 + x_5 + x_6)^3\right) + \epsilon$ | $\{(1,3),(2),(4,5,6)\}$ |
| M4 | $y = x_1 \cdot x_2 + \sin((x_3 + x_4) \cdot \pi) + \log(x_5 \cdot x_6 + 10) + \epsilon$ | $\{(1,2),(3,4),(5,6)\}$ |
| M5 | $y = \exp\left\{\sqrt{x_1^2 + x_2^2 + x_3^2 + x_4^2 + x_5^2 + x_6^2}\right\} + \epsilon$ | $\{(1,2,3,4,5,6)\}$ |

Table 1: Selected models for the simulation study using the exhaustive search method and the corresponding additive group structures.

*Problem (7), then for any $\epsilon > 0$, we have*

$$P\left(|\widehat{\mathcal{R}}(\hat{f}_G) - \mathcal{R}(f_{G^*}^*)| > \epsilon\right) \leq 12n \cdot \exp\left\{\sum_{\boldsymbol{u} \in G} \ln \mathcal{N}\left(\frac{\epsilon}{12|G|}, \mathcal{H}_{\boldsymbol{u}}, d_\infty\right) - \frac{\epsilon^2 n}{144}\right\} +$$

$$12n \cdot \exp\left\{\sum_{\boldsymbol{u} \in G} \ln \mathcal{N}\left(\frac{\epsilon}{12|G|}, \mathcal{H}_{\boldsymbol{u}}, d_\infty\right) - n\left(\frac{\epsilon}{24} - \frac{\lambda_n \|f_{G^*}^*\|^2}{12}\right)^2\right\}. \quad (9)$$

Note that $\lambda_n$ in (9) must be chosen such that $\epsilon/24 - \lambda_n \|f_{G^*}^*\|^2/12$ is positive. For a fixed $p$, the number of amiable group additive structures is finite. Using a Bonferroni type of technique, we can in fact obtain a uniform upper bound for all of the amiable group additive structures in $\mathcal{G}^a$.

**Theorem 2.** *Let $\mathcal{G}^a$ be the set of all amiable group structures. For any $\epsilon > 0$ and $n > 2/\epsilon^2$, we have*

$$P\left(\sup_{G \in \mathcal{G}^a} |\widehat{\mathcal{R}}_g(\hat{f}_G^\lambda) - \mathcal{R}_g(f_{G^*}^*)| > \epsilon\right) \leq 12n|\mathcal{G}^a| \cdot \left[\exp\left\{\max_{G \in \mathcal{G}^a} \ln \mathcal{N}\left(\frac{\epsilon}{12}, \mathcal{H}_G, d_\infty\right) - \frac{\epsilon^2 n}{144}\right\}\right.$$

$$\left. + \exp\left\{\max_{G \in \mathcal{G}^a} \ln \mathcal{N}\left(\frac{\epsilon}{12}, \mathcal{H}_G, d_\infty\right) - n\left(\frac{\epsilon}{24} - \frac{\lambda_n \|f_{G^*}^*\|^2}{12}\right)^2\right\}\right] \quad (10)$$

Next we consider a non-amiable group additive structure $G' \in \mathcal{G} \setminus \mathcal{G}^a$. It turns out that $\hat{\mathcal{R}}(\hat{f}_G)$ fails to converge to $\mathcal{R}(f_{G^*}^*)$, and $|\hat{\mathcal{R}}(\hat{f}_G) - \mathcal{R}(f_{G^*}^*)|$ converges to a positive constant. Furthermore, because the number of non-amiable group additive structures is finite, we can show that $|\hat{\mathcal{R}}(\hat{f}_G) - \mathcal{R}(f_{G^*}^*)|$ is uniformly bounded below from zero with probability going to 1. We state the results below.

**Theorem 3.** *(i) For a non-amiable group structure $G \in \mathcal{G} \setminus \mathcal{G}^a$, there exists a constant $C > 0$ such that $|\hat{\mathcal{R}}_g(\hat{f}_G^\lambda) - \mathcal{R}_g(f_{G^*}^*)|$ converges to $C$ in probability. (ii) There exits a constant $\tilde{C}$ such that $P(|\hat{\mathcal{R}}_g(\hat{f}_G^\lambda) - \mathcal{R}_g(f_{G^*}^*)| > \tilde{C}$ for all $G \in \mathcal{G} \setminus \mathcal{G}^a)$ goes to 1 as $n$ goes to infinity.*

By combining Theorem 2 and Theorem 3, we can prove consistency for our GASI method.

**Theorem 4.** *Let $\lambda_n * n \to 0$. By choosing a proper tuning parameter $\mu > 0$ for the structural penalty , the estimated group structure $\hat{G}$ is consistent for the intrinsic group additive structure $G^*$, that is, $P(\hat{G} = G^*)$ goes to one as the sample size $n$ goes to infinity.*

## 4  Simulation

In this section, we evaluate the performance of GASI using synthetic data. Table 1 lists the five models we are using. Observations of $\boldsymbol{X}$ are simulated independently from $N(0,1)$ in M1, $\text{Unif}(-1,1)$ in M2 and M3, and $\text{Unif}(0,2)$ in M4 and M5. The noise $\epsilon$ is i.i.d. $N(0, 0.01^2)$. The grid values of $\mu$ are equally spaced in $[1e{-}10, 1/64]$ on a log-scale and each $\alpha$ is an integer in $[1, 10]$.

We first show that GASI has the ability to identify the intrinsic group additive structure. The two-step procedure is carried out for each $(\mu, \alpha)$ pair multiple times. If there are $(\mu, \alpha)$ pairs for each model that the true group structure can be often identified, then GASI has the power to identify true group structures. We also apply Algorithm 1 which uses an additional validation set to select the parameters. We simulate 100 different samples for each model. The frequency of the true group structure being identified is calculated for each $(\mu, \alpha)$.

| Model | Max freq. | $\mu$ | $\alpha$ | Max freq. | $\mu$ | $\alpha$ | Max freq. | $\mu$ | $\alpha$ |
|-------|-----------|-------|----------|-----------|-------|----------|-----------|-------|----------|
| M1 | 100 | 1.2500e-06 | 10 | 59 | 1.2500e-06 | 4 | 99 | 1.5625e-02 | 10 |
| M2 | 97 | 1.2500e-06 | 8 | 89 | 1.2500e-06 | 7 | 70 | 1.3975e-04 | 9 |
| M3 | 97 | 1.2500e-06 | 9 | 89 | 1.2500e-06 | 7 | 65 | 1.3975e-04 | 8 |
| M4 | 100 | 1.2500e-06 | 7 | 99 | 1.2500e-06 | 4 | 1 | 1.3975e-04 | 8 |
| M5 | 100 | 1.2500e-06 | 1 | 100 | 1.2500e-06 | 1 | 100 | 1.2500e-06 | 1 |

Table 2: Maximum frequencies that the intrinsic group additive structures are identified for the five models using exhaustive search algorithm without parameter tuning (left panel), with parameter tuning (middle panel) and stepwise algorithm (right panel). If different pairs share the same max frequency, a pair is randomly chosen.

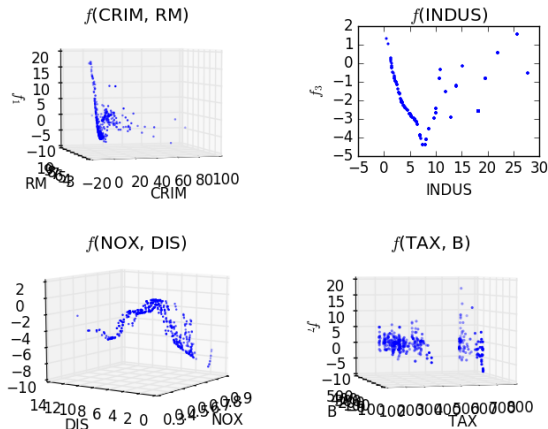

Figure 1: Estimated transformation functions for selected groups. Top-left: group $(1, 6)$, top-right: group $(3)$, bottom-left: group $(5, 8)$, bottom-right: group $(10, 12)$.

In Table 2, we report the maximum frequency and the corresponding $(\mu, \alpha)$ for each model. The complete results are included in the supplementary material. It can be seen from the left panel that the intrinsic group additive structures can be successfully identified. When the parameters are tuned, the middle panel shows that the performance of Model 1 deteriorated. This might be caused by the estimation method (KRR to solve Problem (7)) used in the algorithm. It could also be affected by $\lambda$.

When the number of predictor variables increases, we use a backward stepwise algorithm. We apply Algorithm 2 on the same models. The results are reported in the right panel in Figure 2. The true group structures could be identified most of time for Model 1, 2, 3, 5. The result of Model 4 is not satisfying. Since stepwise algorithm is greedy, it is possible that the true group structures were never visited. Further research is needed to develop a better algorithms.

## 5   Real Data

In this section, we report the results of applying GASI on the Boston Housing data (another real data application is reported in the supplementary material). The data includes 13 predictor variables used to predict the house median value. The sample size is 506. Our goal is to identify a probable group additive structure for the predictor variables. The backward algorithm is used and the tuning parameters $\mu$ and $\alpha$ are selected via 10-fold CV. The group structure that achieves the lowest average validation error is $\{(1, 6), (2, 11), (3), (4, 9), (5, 8), (7, 13), (10, 12)\}$, which is used for further investigation. Then the nonparametric functions for each group were estimated using the whole data set. Because each group contains no more than two variables, the estimated functions can be visualized. Figure 1 shows the selected results.

It is interesting to see some patterns emerging in the plots. For example, the top-left plot shows the function of the average number of rooms per dwelling and per capita crime rate by town. We can see the house value increases with more rooms and decreases as the crime rate increases. However, when the crime rate is low, smaller sized houses (4 or 5 rooms) seem to be preferred. The top-right plot

shows that there is a changing point in terms of how house value is related to the size of non-retail business in the area. The value initially drops when the percentage of non-retail business is small, then increases at around 8%. The increase in the value might be due to the high demand of housing from the employees of those business.

## 6 Discussion

We use group additive model for nonparametric regression and propose a RKHS complexity penalty based approach for identifying the intrinsic group additive structure. There are two main directions for future research. First, our penalty function is based on the covering number of RKHSs. It is of interest to know if there exists other more effective penalty functions. Second, it is of great interest to further improve the proposed method and apply it in general high dimensional nonparametric regression.

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
