[Supplementary Material]

# Group Additive Structure Identification for Kernel Nonparametric Regression

**Pan Chao**
Department of Statistics
Purdue University
West Lafayette, IN 47906
panchao25@gmail.com

**Michael Zhu**
Department of Statistics, Purdue University
West Lafayette, IN 47906
Center for Statistical Science
Department of Industrial Engineering
Tsinghua University, Beijing, China
yuzhu@purdue.edu

## Abstract

The additive model is one of the most popularly used models for high dimensional nonparametric regression analysis. However, its main drawback is that it neglects possible interactions between predictor variables. In this paper, we reexamine the group additive model proposed in the literature, and rigorously define the intrinsic group additive structure for the relationship between the response variable $Y$ and the predictor vector $\boldsymbol{X}$, and further develop an effective structure-penalized kernel method for simultaneous identification of the intrinsic group additive structure and nonparametric function estimation. The method utilizes a novel complexity measure we derive for group additive structures. We show that the proposed method is consistent in identifying the intrinsic group additive structure. Simulation study and real data applications demonstrate the effectiveness of the proposed method as a general tool for high dimensional nonparametric regression.

## 1 Introduction

Regression analysis is popularly used to study the relationship between a response variable $Y$ and a vector of predictor variables $\boldsymbol{X}$. Linear and logistic regression analysis are arguably two most popularly used regression tools in practice, and both postulate explicit parametric models on $f(\boldsymbol{X}) = \mathbb{E}[Y|\boldsymbol{X}]$ as a function of $\boldsymbol{X}$. When no parametric models can be imposed, nonparametric regression analysis can instead be performed. On one hand, nonparametric regression analysis is flexible and not susceptible to model mis-specification, whereas on the other hand, it suffers from a number of well-known drawbacks especially in high dimensional settings. Firstly, the asymptotic error rate of nonparametric regression deteriorates quickly as the dimension of $\boldsymbol{X}$ increases. [23] shows that with some regularity conditions, the optimal asymptotic error rate for estimating a $d$-times differentiable function is $\mathcal{O}\left(n^{-d/(2d+p)}\right)$, where $p$ is the dimensionality of $\boldsymbol{X}$. Secondly, the resulting fitted nonparametric function is often complicated and difficult to interpret.

To overcome the drawbacks of high dimensional nonparametric regression, one popularly used approach is to impose the additive structure [8] on $f(\boldsymbol{X})$, that is to assume that $f(\boldsymbol{X}) = f_1(X_1) + \cdots + f_p(X_p)$ where $f_1, \ldots, f_p$ are $p$ unspecified univariate functions. Thanks to the additive structure, the nonparametric estimation of $f$ or equivalently the individual $f_i$'s for $1 \leq i \leq p$ becomes efficient and does not suffer from the curse of dimensionality. Furthermore, the interpretability of the resulting model has also been much improved.

The key drawback of the additive model is that it does not assume interactions between the predictor variables. To address this limitation, functional ANOVA models were proposed to accommodate higher order interactions, see [7] and [19]. For example, by neglecting interactions of

order higher than 2, the functional ANOVA model can be written as $f(\boldsymbol{X}) = \sum_{i=1}^{p} f_i(X_i) + \sum_{1 \le i,j \le p} f_{ij}(X_i, X_j)$, with some marginal constraints. Another higher order interaction model, $f(\boldsymbol{X}) = \sum_{d=1}^{D} \sum_{1 \le i_1,\dots,i_d \le p} f_j(X_{i_1},\dots,X_{i_d})$, is proposed by [9]. This model considers all interactions of order up to $D$, which is estimated by Kernel Ridge Regression (KRR) [16] with the elementary symmetric polynomial (ESP) kernel.

Both of the two models discussed above assume the existence of possible interactions between any two or more predictor variables. This can lead to a serious problem, that is, the number of nonparametric functions that need to be estimated quickly increases as the number of predictor variables increases. To control the explosion of interaction terms, one approach is to impose the sparsity assumption and then use variable selection methods such as the lasso to select only the important interactions. For the functional ANOVA model, the COSSO method developed by [14] followed this approach. [2] proposes hierarchical kernel learning which assumes that the kernel of inputs is decomposable sum of many basis kernels. Then kernel selection is performed to only select important interactions by imposing group lasso type penalty.

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

 the previous example, we have $G_0$ being the intrinsic group additive structure. If $G^* = G_0$ is known, one only needs to estimate one univariate and two bivariate non-parametric functions. Although $G_1$ and $G_2$ are both amiable, they both require fitting a three-dimensional non-parametric functions. This is both computationally and statistically inefficient. In general, the intrinsic group structure can help much mitigate the curse of dimensionality while improving both efficiency and interpretability of high dimensional nonparametric regression.

## 2.2 Kernel Method with Known Intrinsic Group Additive Structure

Consider $f(\boldsymbol{X}) = \mathbb{E}[Y|\boldsymbol{X}]$. Suppose the intrinsic group additive structure for $f(\boldsymbol{X})$ is known to be $G^* = \{\boldsymbol{u}_j\}_{j=1}^d$, that is, $f(\boldsymbol{X}) = f_{\boldsymbol{u}_1}(\boldsymbol{X}_{\boldsymbol{u}_1}) + \cdots + f_{\boldsymbol{u}_d}(\boldsymbol{X}_{\boldsymbol{u}_d})$. Therefore, estimating $f$ is essentially to estimate the functions, $f_{\boldsymbol{u}_1}, f_{\boldsymbol{u}_2}, \ldots, f_{\boldsymbol{u}_d}$. We will use the kernel method. Let $(K_{\boldsymbol{u}_j}, \mathcal{H}_{\boldsymbol{u}_j})$ be the kernel and its corresponding RKHS for the $j$-th group $\boldsymbol{u}_j$. Then using kernel methods is to solve

$$\hat{f}_{\lambda, G^*} = \argmin_{f_{G^*} \in \mathcal{H}_{G^*}} \left\{ \frac{1}{n} \left( y_i - f_{G^*}(\boldsymbol{x}_i) \right)^2 + \lambda \| f_{G^*} \|_{\mathcal{H}_{G^*}}^2 \right\}, \tag{2}$$

where $\mathcal{H}_{G^*} := \{f = \sum_{j=1}^d f_{\boldsymbol{u}_j} \mid f_{\boldsymbol{u}_j} \in \mathcal{H}_{\boldsymbol{u}_j}\}$. The subscripts are used on RHS to explicitly indicate the dependence of the solution on the group additive structure $G^*$ and tuning parameter $\lambda$. The solution is searched in the corresponding direct sum RKHS and can be solved by KRR.

In general, an RKHS is usually smaller than the $L^2$ space defined on the same input domain. So, it is not always that $\hat{f}_{\lambda, G^*} \equiv f$, and in fact a bias exists. However, one can choose to use kernels $K_{\boldsymbol{u}_j}$ that are universal in the sense that their corresponding RKHSs are dense in the $L^2$ spaces (see [22], [5]). Two examples of universal kernel are Gaussian and Laplace. By using universal kernels, not only can the bias of $\hat{f}_{\lambda, G^*}$ reduces to zero as $n$ goes to infinity, but also can $\hat{f}_{\lambda, G^*}$ recover the structural properties such as the group additive structure of $f(\boldsymbol{X})$. This is the fundamental reason for the consistency property of our proposed method to identify the intrinsic group additive structure.

## 2.3 Identification of Unknown Intrinsic Group Additive Structure

### 2.3.1 Penalization on Group Additive Structures

The success of the kernel method discussed in the previous subsection hinges on the knowledge of the intrinsic group additive structure $G^*$. In practice, however, $G^*$ is seldom known, and it may be of primary interest to identify $G^*$ while fitting a nonparametric regression for $\mathbb{E}[Y|\boldsymbol{X}]$ as discussed earlier. Recall that in Subsection 2.1, we have shown that $G^*$ exists and unique. The other possible group additive structures belong to two categories, amiable and non-amiable.

Let's consider an arbitrary non-amiable group additive structure $G \in \mathcal{G} \setminus \mathcal{G}^a$ first. Suppose $G$ is used in the place of $G^*$ in the kernel method (2). The resulting fitted function $\hat{f}_{\lambda, G}$, as an estimator of $f$, will have a systematic bias because the $L^2$ distance between any function $f_G$ in $\mathcal{H}_G$ and the true function $f$ will be bounded below. The bias remains regardless of the size of the training sample. In other words, using a non-amiable group additive structure will result in poor fitting of the model.

Next we consider an arbitrary amiable group additive structure $G^a$. Suppose $G^a$ is used in the place of $G^*$ in (2). Recall that because $G^a$ is amiable, therefore for the true function $f(\boldsymbol{X})$, we have $f_{G^*} = f_{G^a}$ almost surely.

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

)$, where we explicitly indicate the dependency of $\alpha$ on $\epsilon$. Now we could use the rate as the explicit expression of the complexity measure $\mathcal{C}(G)$ in Problem (3), that is $\mathcal{C}(G) = \sum_{j=1}^{d} \alpha(\epsilon)^{|\boldsymbol{u}_j|+1}$. Recall that there is another tuning parameter $\mu$ which controls the effect of the complexity of group structure has on the penalized risk. By factoring out the common 1 in the exponent for all groups and combining it with $\mu$, we could further simplify the penalty's expression. Thus, we have the following explicit formulation for AGSI which simultaneously solves the non-parametric regression problem.

$$\hat{f}_{\lambda,\mu}, \hat{G} = \underset{f_G \in \mathcal{H}_G, G}{\arg\min} \left\{ \sum_{i=1}^{n} (y_i - f_G(\boldsymbol{x}_i))^2 + \lambda \|f_G\|_{\mathcal{H}_G}^2 + \mu \sum_{j=1}^{d} \alpha^{|\boldsymbol{u}_j|} \right\}. \tag{6}$$

## 2.4 Estimation

We assume that the value of $\lambda$ is pre-specified. In practice, this parameter can be tuned separately. If the values of $\mu$ and $\alpha$ are given, Problem (6) can be solved by following a two-step procedure. First, when the group structure $G$ is given, the functions $f_{\boldsymbol{u}}$ can be estimated by using KRR to solve the following problem

$$\hat{\mathcal{R}}_G^\lambda = \underset{f_G \in \mathcal{H}_G}{\min} \left\{ \frac{1}{n} \sum_{i=1}^{n} (y_i - f_G(\boldsymbol{x}_i))^2 + \lambda \|f_G\|_{\mathcal{H}_G}^2 \right\}. \tag{7}$$

Second, the optimal group structure is chosen to achieve both small fitting error and the complexity penalty, i.e.

$$\hat{G} = \underset{G \in \mathcal{G}}{\arg\min} \left\{ \hat{\mathcal{R}}_G^\lambda + \mu \sum_{j=1}^{d} \alpha^{|\boldsymbol{u}_j|} \right\}. \tag{8}$$

The two-step procedure above is expected to identify the intrinsic group structure, that is, $\hat{G} = G^*$. Recall a group structure belongs to one of the three categories, intrinsic, amiable, or non-amiable structures. If $G$ is non-amiable, then $\hat{\mathcal{R}}_G^\lambda$ is expected to be large, because $G$ is a wrong structure and will result in a biased estimate. If $G$ is amiable, though $\hat{\mathcal{R}}_G^\lambda$ is expected to be small, the complexity penalty of $G$ is larger than that for $G^*$. As a consequence, only $G^*$ can simultaneously achieve a small $\hat{\mathcal{R}}_{G^*}^\lambda$ and a relatively small complexity penalty. Therefore, when the sample size is large enough, we expect $\hat{G} = G^*$ with high probability. If the values of the turning parameters $\mu$ and $\alpha$ are not given, a separate validation set can be used to select tuning parameters. The estimation is summarized in Algorithm 1

| Algorithm 1: Exhaustive Search w/ Validation | Algorithm 2: Basic Backward Stepwise |
|---|---|
| 1: Split data into training ($\mathcal{T}$) and validation ($\mathcal{V}$) sets. | 1: State with the group structure $\{(1,\ldots,p)\}$; |
| 2: **for** $(\mu, \alpha)$ in grid **do** | 2: Solve (6) and obtain its minimum value $\hat{\mathcal{R}}_G^{\text{pen}}$; |
| 3:     **for** $G \in \mathcal{G}$ **do** | 3: **for** each predictor variable $j$ **do** |
| 4:        $\hat{\mathcal{R}}_G, \hat{f}_G \leftarrow$ solve (7) using $G$; | 4:     $G' \leftarrow$ either split $j$ as a new group or add to an existing group; |
| 5:        Calculate the sum in (8), denoted by $\hat{\mathcal{R}}_G^{\text{pen},\mu,\alpha}$; | 5:     Solve (6) and obtain its minimum value $\hat{\mathcal{R}}_{G'}^{\text{pen}}$; |
| 6:     **end for** | 6:     **if** $\hat{\mathcal{R}}_{G'}^{\text{pen}} < \hat{\mathcal{R}}_G^{\text{pen}}$ **then** |
| 7:     $\hat{G}^{\mu,\alpha} \leftarrow \arg\min_{G \in \mathcal{G}} \hat{\mathcal{R}}_G^{\text{pen},\mu,\alpha}$; | 7:        Keep $G'$ as the new group structure; |
| 8:     $\hat{y}^{\mathcal{V}} \leftarrow \hat{f}_{\hat{G}^{\mu,\alpha}}(\boldsymbol{x}^{\mathcal{V}})$; | 8:     **end if** |
| 9:     $e^2_{\hat{G}^{\mu,\alpha}} \leftarrow \|y^{\mathcal{V}} - \hat{y}^{\mathcal{V}}\|^2$; | 9: **end for** |
| 10: **end for** | 10: **return** $G'$; |
| 11: $\mu^*, \alpha^* \leftarrow \arg\min_{\mu,\alpha} e^2_{\hat{G}^{\mu,\alpha}}$; | |
| 12: $G^* \leftarrow \hat{G}^{\mu^*,\alpha^*}$; | |

Algorithm 1 selects the group additive structure by compare the results of all possible group structures. When a model contains a large number of predictor variables, such exhaustive search suffers high computational cost. In order to apply GASI on a large model, we propose a backward stepwise algorithm which is illustrated in Algorithm 2.

## 3 Theory

In this section, we prove that the estimated group additive structure $\hat{G}$ as a solution to (6) is consistent, that is the probability $P(\hat{G} = G^*)$ goes to 1 as the sample size $n$ goes to infinity. As we discussed before, when a non-amiable group additive structure is used, the solution of a usual kernel nonparametric regression problem has a non-zero bias. While all amiable group additive structures give unbiased estimates, using the intrinsic group additive structure will enjoy the fastest rate of convergence. Thus, the new complexity penalty is used to filter out all amiable group structures with slow convergence rate. We provide the main theorems in this section. The proof and supporting lemmas are included in the supplemental document.

Let $\mathcal{R}(f_G) := \mathbb{E}[(Y - f(\boldsymbol{X}))^2]$ denote the population risk of function $f \in \mathcal{H}_G$, and $\hat{\mathcal{R}}(f) := \frac{1}{n} \sum_{i=1}^{n}(y_i - f(\boldsymbol{x}_i))^2$ be the empirical risk. First, we show that for any given amiable group additive structure $G \in \mathcal{G}^a$, its optimized empirical risk $\hat{\mathcal{R}}(\hat{f}_G)$ converges in probability to the optimal population risk $\mathcal{R}(f_{G^*}^*)$ achieved by the intrinsic group additive structure. Here $\hat{f}_G$ denotes the minimizer of the regularized empirical risk (7) when the group additive structure $G$ is used, and $f_{G^*}^*$ denotes the minimizer of the population risk when the intrinsic group structure is used. The result is given below as Proposition 4.

**Proposition 4.** *Let $G^*$ be the intrinsic group additive structure, $G \in \mathcal{G}^a$ a given amiable group structure, and $\mathcal{H}_{G^*}$ and $\mathcal{H}_G$ the respective direct sum RKHSs. If $\hat{f}_G^{\lambda} \in \mathcal{H}_G$ is the optimal solution of Problem (7), then for any $\epsilon > 0$, we have*

$$P\left(|\hat{\mathcal{R}}(\hat{f}_G) - \mathcal{R}(f_{G^*}^*)| > \epsilon\right) \le 12n \cdot \exp\left\{\sum_{\boldsymbol{u} \in G} \ln \mathcal{N}\left(\frac{\epsilon}{12|G|}, \mathcal{H}_{\boldsymbol{u}}, d_\infty\right) - \frac{\epsilon^2 n}{144}\right\} + $$

$$12n \cdot \exp\left\{\sum_{\boldsymbol{u} \in G} \ln \mathcal{N}\left(\frac{\epsilon}{12|G|}, \mathcal{H}_{\boldsymbol{u}}, d_\infty\right) - n\left(\frac{\epsilon}{24} - \frac{\lambda_n \|f_{G^*}^*\|^2}{12}\right)^2\right\}. \quad (9)$$

Note that $\lambda_n$ in (21) must be chosen such that $\frac{\epsilon}{24} - \frac{\lambda_n \|f_{G^*}^*\|^2}{12}$ is positive. For any given $\epsilon$, when $n$ is sufficiently large, the exponents of the two terms in (21) will become negative. When $n$ further increases, both of the terms in (21) will decrease exponentially to zero. Therefore, Proposition 4 implies that $\hat{\mathcal{R}}(\hat{f}_G)$ converges to $\mathcal{R}(f_{G^*}^*)$ in probability. For a fixed $p$ and intrinsic group additive structure, the number of amiable group additive structures is finite. Using a Bonferroni type of

| ID | Model | Intrinsic Group Structure |
|---|---|---|
| M1 | $y = 2x_1 + x_2^2 + x_3^3 + \sin(\pi x_4) + \log(x_5 + 5) + |x_6| + \epsilon$ | $\{(1),(2),(3),(4),(5),(6)\}$ |
| M2 | $y = \frac{1}{1+x_1^2} + \arcsin\left(\frac{x_2+x_3}{2}\right) + \arctan\left((x_4 + x_5 + x_6)^3\right) + \epsilon$ | $\{(1),(2,3),(4,5,6)\}$ |
| M3 | $y = \arcsin\left(\frac{x_1+x_3}{2}\right) + \frac{1}{1+x_2^2} + \arctan\left((x_4 + x_5 + x_6)^3\right) + \epsilon$ | $\{(1,3),(2),(4,5,6)\}$ |
| M4 | $y = x_1 \cdot x_2 + \sin((x_3 + x_4) \cdot \pi) + \log(x_5 \cdot x_6 + 10) + \epsilon$ | $\{(1,2),(3,4),(5,6)\}$ |
| M5 | $y = \exp\left\{\sqrt{x_1^2 + x_2^2 + x_3^2 + x_4^2 + x_5^2 + x_6^2}\right\} + \epsilon$ | $\{(1,2,3,4,5,6)\}$ |

Table 1: Selected models for the simulation study using the exhaustive search method and the corresponding additive group structures.

technique, we can in fact obtain a uniform upper bound for all of the amiable group additive structures in $\mathcal{G}^a$. This result is stated in the following theorem.

**Theorem 2.** *Let $\mathcal{G}^a$ be the set of all amiable group structures. For any $\epsilon > 0$ and $n > 2/\epsilon^2$, we have*

$$P\left(\sup_{G \in \mathcal{G}^a} |\widehat{\mathcal{R}}_g(\hat{f}_G^\lambda) - \mathcal{R}_g(f_{G^*}^*)| > \epsilon\right) \leq 12n|\mathcal{G}^a| \cdot \left[\exp\left\{\max_{G \in \mathcal{G}^a} \ln \mathcal{N}\left(\frac{\epsilon}{12}, \mathcal{H}_G, d_\infty\right) - \frac{\epsilon^2 n}{144}\right\}\right.$$

$$\left. + \exp\left\{\max_{G \in \mathcal{G}^a} \ln \mathcal{N}\left(\frac{\epsilon}{12}, \mathcal{H}_G, d_\infty\right) - n\left(\frac{\epsilon}{24} - \frac{\lambda_n \|f_{G^*}^*\|^2}{12}\right)^2\right\}\right] \tag{10}$$

Theorem 2 implies that the convergence of $\widehat{\mathcal{R}}(\hat{f}_G)$ to $\mathcal{R}(f_{G^*}^*)$ in probability is uniform for $G$ in $\mathcal{G}^a$.

Next we consider a non-amiable group additive structure $G' \in \mathcal{G} \setminus \mathcal{G}^a$. It turns out that $\widehat{\mathcal{R}}(\hat{f}_G)$ fails to converge to $\mathcal{R}(f_{G^*}^*)$ in probability, and $|\widehat{\mathcal{R}}(\hat{f}_G) - \mathcal{R}(f_{G^*}^*)|$ converges to a positive constant in probability. Furthermore, because the number of non-amiable group additive structures is finite, we can show that $|\widehat{\mathcal{R}}(\hat{f}_G) - \mathcal{R}(f_{G^*}^*)|$ is uniformly bounded below from zero with probability going to 1. We state the results as the next theorem.

**Theorem 3.** *(i) For a non-amiable group structure $G' \in \mathcal{G} \setminus \mathcal{G}^a$, there exists a constant $C > 0$ such that $|\widehat{\mathcal{R}}_g(\hat{f}_{G'}^\lambda) - \mathcal{R}_g(f_{G^*}^*)|$ converges to $C$ in probability.*
*(ii) There exits a constant $\tilde{C}$ such that $P(|\widehat{\mathcal{R}}_g(\hat{f}_{G'}^\lambda) - \mathcal{R}_g(f_{G^*}^*)| > \tilde{C}$ for all $G' \in \mathcal{G} \setminus \mathcal{G}^a)$ goes to 1 as $n$ goes to infinity.*

By combining Theorem 2 and Theorem 3, it is not difficult to show that the probability that the solution of (6) $\hat{G}$ is not equal to the intrinsic group additive structure goes to zero as $n$ goes to infinity. The structural penalty helps to distinguish amiable structures from the intrinsic group additive structure. We state this result in the following theorem.

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

The second data set is the communities and crime data (unnormalized). It combines socio-economic, law enforcement, and crime data collected by US government agencies. There are 2215 samples and 147 variables with missing values. We choose Number of Murders in 1995 to be the response in this study and investigate its relationship between the predictor variables. We removed the observations with missing values.

To deal with the large number of predictor variables, a screening procedure is used to select the most related variables. We fit OKGT for each of the 122 predictor against the response, then keep the variables with $R^2 > 0.99$. This ensures that the selected predictors are highly dependent to the response. The screening procedure selects 23 predictor variables. We also remove the samples with

Figure 1: Estimated transformation functions for selected groups. Top-left: group $(1,6)$, top-right: group $(3)$, bottom-left: group $(5,8)$, bottom-right: group $(10,12)$.

Figure 2: Selected results for the communities and crime data where the number of murders is the response. The blue dots are the transformed observation of the predictor variable. The red line is the estimated function.

missing values, which reduced sample size to 343. Then the backward algorithm is applied on the prepared data set.

The procedure selected the fully additive group structure. Figure 7 includes selected results which show highly nonlinear effect of each predictor variable. The first plot shows that the effect of Median Family Income is almost zero until it reaches the high end where murders drop dramatically. The second plot shows an interesting pattern for Total Requests for Police per Police Officer. As the number of requests increases, the number of murders initially decreases slowly. One reason for this is that increasing requests cause more presence of police in the area which is helpful to control crimes. However, murders increase quickly as the number of requests enters the high range. An explanation for this is that the surging number of requests for police is due to the low security and high murder rate in the area.

## 6 Discussion

We use group additive model for nonparametric regression and propose a RKHS complexity penalty based approach for identifying the intrinsic group additive structure. There are two main directions for future research. First, our penalty is based on the covering number of RKHSs. It is of interest to know if there exist other more effective penalty. Second, the current backward stepwise algorithm may become unstable and fail to achieve the potential in identifying the true additive group structure. It is of great interest to further improve the proposed method that can be applied in general high dimensional nonparametric regression.

## Footnotes

[1] If $G'$ is amiable, then a subset $\boldsymbol{u}$ of $G'$ always assumes an additive structure. So there is no error between $f^*_{G^*}$ and $\hat{f}_{G'}$ after such a decomposition.

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

# Appendix

## A. Theorem, Proposition, Lemma and Proof

**Proposition 1.** *Let $G_1$ and $G_2$ be two group additive structures. If $G_1 \leq G_2$, then $L^2_{G_1} \subseteq L^2_{G_2}$. Furthermore, if $X_1, \ldots, X_p$ are independent and $G_1 \neq G_2$, then $L^2_{G_1} \subset L^2_{G_2}$.*

*Proof.* First we prove the first part.

Since $f \in L^2_{G_1}$, we have $f = \sum_{\boldsymbol{u} \in G_1} f_{\boldsymbol{u}}(\boldsymbol{x_u})$.

If $G_1 \cap G_2 \neq \emptyset$, then for each $\boldsymbol{u} \in G_1 \cap G_2$, it is true that $f_{\boldsymbol{u}} \in L^2_{G_2}$.

If $\boldsymbol{u} \notin G_1 \cap G_2$ and $\boldsymbol{u} \in G_1 \setminus G_2$, because $G_1 \leq G_2$, there exists $\boldsymbol{u}_1, \ldots, \boldsymbol{u}_k \in G_2 \setminus G_1$ for some $k < |G_2|$ such that $\boldsymbol{v} := \boldsymbol{u} \cup \boldsymbol{u}_1 \cup \cdots \cup \boldsymbol{u}_k \in G_2$. Since

$$L^2([0,1]^{|\boldsymbol{u}|}) \oplus L^2([0,1]^{|\boldsymbol{u}_1|}) \oplus \cdots \oplus L^2([0,1]^{|\boldsymbol{u}_k|}) \subseteq L^2([0,1]^{|\boldsymbol{v}|}), \tag{11}$$

by induction, we have the desired result.

The sub-additivity in (11) is true because for two groups $\boldsymbol{u}$ and $\boldsymbol{v}$ in a group structure $G$, we have

$$
\int (f_{\boldsymbol{u}}(\boldsymbol{x_u}) + f_{\boldsymbol{v}}(\boldsymbol{x_v}))^2 p(\boldsymbol{x_u}, \boldsymbol{x_v}) d\boldsymbol{x_u} d\boldsymbol{x_v}
$$

$$
= \int f_{\boldsymbol{u}}^2(\boldsymbol{x_u}) p(\boldsymbol{x_u}, \boldsymbol{x_v}) d\boldsymbol{x_u} d\boldsymbol{x_v} + \int f_{\boldsymbol{v}}^2(\boldsymbol{x_v}) p(\boldsymbol{x_u}, \boldsymbol{x_v}) d\boldsymbol{x_u} d\boldsymbol{x_v}
$$

$$
+ 2 \int f_{\boldsymbol{u}}(\boldsymbol{x_u}) f_{\boldsymbol{v}}(\boldsymbol{x_v}) p(\boldsymbol{x_u}, \boldsymbol{x_v}) d\boldsymbol{x_u} d\boldsymbol{x_v}
$$

$$
\leq \int f_{\boldsymbol{u}}^2(\boldsymbol{x_u}) p(\boldsymbol{x_u}, \boldsymbol{x_v}) d\boldsymbol{x_u} d\boldsymbol{x_v} + \int f_{\boldsymbol{v}}^2(\boldsymbol{x_v}) p(\boldsymbol{x_u}, \boldsymbol{x_v}) d\boldsymbol{x_u} d\boldsymbol{x_v}
$$

$$
+ 2 \left( \int f_{\boldsymbol{u}}^2(\boldsymbol{x_u}) p(\boldsymbol{x_u}, \boldsymbol{x_v}) d\boldsymbol{x_u} d\boldsymbol{x_v} \right)^{1/2} \cdot \left( \int f_{\boldsymbol{v}}^2(\boldsymbol{x_v}) p(\boldsymbol{x_u}, \boldsymbol{x_v}) d\boldsymbol{x_u} d\boldsymbol{x_v} \right)^{1/2}
$$

$$
< \infty
$$

The second to the last inequality is due to Holder's inequality with $p = q = 2$.

We further need to show the proper part (i.e. strict subset).

For $\boldsymbol{u}, \boldsymbol{v} \in G_1$, $\boldsymbol{u}, \boldsymbol{v} \notin G_2$, $\boldsymbol{u} \cup \boldsymbol{v} \in G_2$, we need to show that there is a function $h(\boldsymbol{x_u}, \boldsymbol{x_v}) \in L^2([0,1]^{|\boldsymbol{u} \cup \boldsymbol{v}|})$ which does not belong to $L^2_{\boldsymbol{u}} \oplus L^2_{\boldsymbol{v}}$. That is

$$\inf_{\substack{f \in L^2_{\boldsymbol{u}} \\ g \in L^2_{\boldsymbol{v}}}} \int (h(\boldsymbol{x_u}, \boldsymbol{x_v}) - f(\boldsymbol{x_u}) - g(\boldsymbol{x_v}))^2 p(\boldsymbol{x_u}, \boldsymbol{x_v}) d\boldsymbol{x_u} d\boldsymbol{x_v} > 0 \tag{12}$$

Define the following functional of $f$ and $g$ as

$$F(f,g) := \int (h(\boldsymbol{x_u}, \boldsymbol{x_v}) - f(\boldsymbol{x_u}) - g(\boldsymbol{x_v}))^2 p(\boldsymbol{x_u}, \boldsymbol{x_v}) d\boldsymbol{x_u} d\boldsymbol{x_v} \tag{13}$$

Let $\delta(\boldsymbol{x_u})$ be the Gâteaux's derivative at $\boldsymbol{x_u}$, then

$$F(f_{\boldsymbol{u}} + t\delta_{\boldsymbol{u}}, g_{\boldsymbol{v}}) - F(f_{\boldsymbol{u}}, g_{\boldsymbol{v}}) = \int \left( 2tf\delta + t^2 \delta^2 - 2th\delta + 2tg\delta \right) p_{\boldsymbol{uv}} d\boldsymbol{x_u} d\boldsymbol{x_v}$$

At minimum, we have

$$
\lim_{t \to 0} \frac{F(f_{\boldsymbol{u}} + t\delta_{\boldsymbol{u}}, g_{\boldsymbol{v}}) - F(f_{\boldsymbol{u}}, g_{\boldsymbol{v}})}{t}
$$

$$
= \lim_{t \to 0} \int \left( 2f\delta + t\delta^2 - 2h\delta + 2g\delta \right) p_{\boldsymbol{uv}} d\boldsymbol{x_u} d\boldsymbol{x_v}
$$

$$
= \int \left( 2f\delta - 2h\delta + 2g\delta \right) p_{\boldsymbol{uv}} d\boldsymbol{x_u} d\boldsymbol{x_v}
$$

$$
= 0 \tag{14}
$$

Since (14) holds for all $\delta \in L_{\boldsymbol{u}}^2$, then we have

$$
\int \left( f + g - h \right) p_{\boldsymbol{uv}} d\boldsymbol{x_v} = 0 \tag{15}
$$

By symmetry, we also have the following identity.

$$
\int \left( f + g - h \right) p_{\boldsymbol{uv}} d\boldsymbol{x_u} = 0 \tag{16}
$$

Since $h$ is given, we set $C_1 = \int h p_{\boldsymbol{uv}} dv$ and $C_2 = \int h p_{\boldsymbol{uv}} du$.

Solving (15) for $f$ we have,

$$
f = \frac{\int h p_{\boldsymbol{uv}} d\boldsymbol{x_v} - \int g p_{\boldsymbol{uv}} d\boldsymbol{x} v}{\int p_{\boldsymbol{uv}} d\boldsymbol{x_v}} \tag{17}
$$

Plug (17) into (16), we have

$$
\int \frac{\int h p_{\boldsymbol{uv}} d\boldsymbol{x_v} - \int g p_{\boldsymbol{uv}} d\boldsymbol{x_v}}{\int p_{\boldsymbol{uv}} d\boldsymbol{x_v}} p_{\boldsymbol{uv}} d\boldsymbol{x_u} + g \int p_{\boldsymbol{uv}} d\boldsymbol{x_u} - \int h p_{\boldsymbol{uv}} d\boldsymbol{x_u} = 0
$$

$$
\Leftrightarrow \quad g \int p_{\boldsymbol{uv}} d\boldsymbol{x_u} - \int \frac{\int g p_{\boldsymbol{uv}} d\boldsymbol{x_v}}{\int p_{\boldsymbol{uv}} d\boldsymbol{x_v}} p_{\boldsymbol{uv}} d\boldsymbol{x_u} = \int h p_{\boldsymbol{uv}} d\boldsymbol{x_u} - \int \frac{\int h p_{\boldsymbol{uv}} d\boldsymbol{x_v}}{\int p_{\boldsymbol{uv}} d\boldsymbol{x_v}} p_{\boldsymbol{uv}} d\boldsymbol{x_u} \tag{18}
$$

Since $\boldsymbol{X_u} \perp \boldsymbol{X_v}$, we have $p_{\boldsymbol{uv}} = p_{\boldsymbol{u}} p_{\boldsymbol{v}}$. Then, identity (18) is equivalent to

$$
g - \int g p_{\boldsymbol{v}} d\boldsymbol{x_v} = \int h p_{\boldsymbol{u}} d\boldsymbol{x_u} - \int \int h p_{\boldsymbol{v}} d\boldsymbol{x_v} p_{\boldsymbol{u}} d\boldsymbol{x_u}.
$$

This is a Fredholm integral equation, with solution

$$
\begin{cases} f = \int h p_{\boldsymbol{v}} d\boldsymbol{x_v} - C \\ g = \int h p_{\boldsymbol{u}} d\boldsymbol{x_u} + C \end{cases} \tag{19}
$$

where $C$ is any constant.

To this end, the minimum approximation error in (12) achieves 0 when the following identity is true almost surely.

$$
h = \int h p_{\boldsymbol{v}} d\boldsymbol{x_v} + \int h p_{\boldsymbol{u}} d\boldsymbol{x_u}
$$

A counter example is given by $h(\boldsymbol{x_u}, \boldsymbol{x_v}) = \sin(\boldsymbol{x_u} + \boldsymbol{x_v})$ which does not assume the above decomposition. So $L_{\boldsymbol{u}}^2 \oplus L_{\boldsymbol{v}}^2$ is a proper subspace of $h(\boldsymbol{x_u}, \boldsymbol{x_v}) \in L^2([0,1]^{|\boldsymbol{u} \cup \boldsymbol{v}|})$.

Thus the proposition is proved. $\qquad \square$

**Theorem 1.** *Let $\mathcal{G}^a$ be the set of amiable group additive structures for $f$. There is a unique minimal group additive structure $G^* \in \mathcal{G}^a$ such that $G^* \leq G$ for all $G \in \mathcal{G}^a$, where the order is given by Definition 1. $G^*$ is called the **intrinsic group additive structure** for $f$.*

*Proof.* Since the partial order is defined for any subset of group structures in $\mathcal{G}^a$, the existence of $G^*$ is the result of Zorn's Lemma. The uniqueness is due to the fact that $\mathcal{G}^a$ is a finite set. $\qquad\square$

**Proposition 3.** *Let $G$ be a group additive structure and $\mathcal{H}_G$ be the induced direct sum RKHS defined in (3). Then, we have the following inequality relating the covering number of $\mathcal{H}_G$ and the covering numbers of $\mathcal{H}_{\boldsymbol{u}_j}$*

$$\ln\mathcal{N}\left(\epsilon, I_G, d_\infty\right) \leq \sum_{j=1}^{d} \ln\mathcal{N}\left(\frac{\epsilon}{|G|}, I_{\boldsymbol{u}_j}, d_\infty\right), \tag{20}$$

*where $|G|$ denotes the number of groups in $G$.*

*Proof.* Due to Lemma 1, we have $\mathcal{N}\left(\epsilon, I_G, d_\infty\right) \leq \Pi_{\boldsymbol{u}\in G}\mathcal{N}\left(\frac{\epsilon}{|G|}, \overline{I(\widetilde{\mathcal{H}}_{\boldsymbol{u}})}, d_\infty\right) = \Pi_{\boldsymbol{u}\in G}\mathcal{N}\left(\frac{\epsilon}{|G|}, I_{\boldsymbol{u}}, d_\infty\right)$. Then, taking log on both sides gives the desired result. $\qquad\square$

**Proposition 4.** *Let $G^*$ be the intrinsic group additive structure, $G \in \mathcal{G}^a$ a given amiable group structure, and $\mathcal{H}_{G^*}$ and $\mathcal{H}_G$ the respective direct sum RKHSs. If $\hat{f}_G^\lambda \in \mathcal{H}_G$ is the optimal solution of Problem (7), then for any $\epsilon > 0$, we have*

$$P\left(|\widehat{\mathcal{R}}(\hat{f}_G) - \mathcal{R}(f_{G^*}^*)| > \epsilon\right) \leq 12n \cdot \exp\left\{\sum_{\boldsymbol{u}\in G}\ln\mathcal{N}\left(\frac{\epsilon}{12|G|}, \mathcal{H}_{\boldsymbol{u}}, d_\infty\right) - \frac{\epsilon^2 n}{144}\right\} + $$

$$12n \cdot \exp\left\{\sum_{\boldsymbol{u}\in G}\ln\mathcal{N}\left(\frac{\epsilon}{12|G|}, \mathcal{H}_{\boldsymbol{u}}, d_\infty\right) - n\left(\frac{\epsilon}{24} - \frac{\lambda_n\|f_{G^*}^*\|^2}{12}\right)^2\right\}. \tag{21}$$

*Proof.* Since the following inequality holds,

$$|\widehat{\mathcal{R}}_g(\hat{f}_G) - \mathcal{R}_g(f_{G^*}^*)| \leq |\widehat{\mathcal{R}}_g(\hat{f}_G) - \mathcal{R}_g(\hat{f}_G)| + |\mathcal{R}_g(\hat{f}_G) - \mathcal{R}_g(f_{G^*}^*)|, \tag{22}$$

the upper bound for the desired deviation can be derived from the upper bounds of the two terms on RHS in the inequality.

The upper bound for the first term can be derived by using the uniform convergence bound in [1] (also see Lemma 12.38 in [20]). So we have the following probabilistic upper bound for the first term. For all $n > \frac{8}{\epsilon^2}$,

$$P\left(|\widehat{\mathcal{R}}_g(\hat{f}_G) - \mathcal{R}_g(\hat{f}_G)| > \frac{\epsilon}{2}\right)$$

$$\leq 12n \cdot \mathbb{E}\left[\mathcal{N}\left(\frac{\epsilon}{12}, \mathcal{H}_G, \ell_\infty^{X'}\right)\right] \cdot \exp\left\{-\frac{\epsilon^2 n}{144}\right\}$$

$$\leq 12n \cdot \exp\left\{\ln\mathcal{N}^{(n)}\left(\frac{\epsilon}{12}, \mathcal{H}_G, \ell_\infty\right) - \frac{\epsilon^2 n}{144}\right\}$$

$$\leq 12n \cdot \exp\left\{\ln\mathcal{N}\left(\frac{\epsilon}{12}, \mathcal{H}_G, \ell_\infty\right) - \frac{\epsilon^2 n}{144}\right\}, \tag{23}$$

where $\ell_\infty^{X'}$ denotes the sup-norm of function $f \in \mathcal{F}$ restricted to the sample $X' = \{x_1', \ldots, x_n'\}$ which is independent of the sample $X = \{x_1, \ldots, x_n\}$ used for estimation and $\mathcal{N}^{(n)}\left(\epsilon, \mathcal{H}, \ell_\infty\right)$ is called the $\epsilon$-*growth function* of the space $\mathcal{H}$ which is defined as

$$\mathcal{N}^{(n)}\left(\epsilon, \mathcal{H}, \ell_\infty\right) := \sup_{\boldsymbol{x}_1, \ldots, \boldsymbol{x}_n \in \mathcal{X}} \mathcal{N}\left(\epsilon, \mathcal{H}, \ell_\infty^X\right).$$

The second inequality is due to the fact that $\mathbb{E}\left[\mathcal{N}\left(\epsilon, \mathcal{H}, \ell_\infty^{X'}\right)\right] \leq \mathcal{N}^{(n)}\left(\epsilon, \mathcal{H}\right)$.

The upper bound for the second term in 22 can be derived by repeatedly applying the same uniform convergence bound. Due to Lemma 2, we have for all $\epsilon > 0$ and all $n > 2/\epsilon^2$,

$$P\left(|\mathcal{R}_g(\hat{f}_G) - \mathcal{R}_g(f_{G^*}^*)| > \frac{\epsilon}{2}\right)$$

$$\leq 12n \cdot \ln \mathbb{E}\left[\mathcal{N}\left(\frac{\epsilon^2}{12}, \mathcal{H}_G, \ell_\infty^{X'}\right)\right] \cdot \exp\left\{-n\left(\frac{\epsilon}{24} - \frac{\lambda_n\|f_{G^*}^*\|^2}{12}\right)^2\right\}$$

$$\leq 12n \cdot \exp\left\{\ln \mathcal{N}^{(n)}\left(\frac{\epsilon}{12}, \mathcal{H}_G, \ell_\infty\right) - n\left(\frac{\epsilon}{24} - \frac{\lambda_n\|f_{G^*}^*\|^2}{12}\right)^2\right\}$$

$$\leq 12n \cdot \exp\left\{\ln \mathcal{N}\left(\frac{\epsilon}{12}, \mathcal{H}_G, \ell_\infty\right) - n\left(\frac{\epsilon}{24} - \frac{\lambda_n\|f_{G^*}^*\|^2}{12}\right)^2\right\}. \tag{24}$$

By plugging the upper bounds (23) and (24) in (22), we have

$$P\left(|\widehat{\mathcal{R}}_g(\hat{f}_G) - \mathcal{R}_g(f_{G^*}^*)| > \epsilon\right)$$

$$\leq 12n \cdot \exp\left\{\ln \mathcal{N}\left(\frac{\epsilon}{12}, \mathcal{H}_G, \ell_\infty\right) - \frac{\epsilon^2 n}{144}\right\} +$$

$$12n \cdot \exp\left\{\ln \mathcal{N}\left(\frac{\epsilon}{12}, \mathcal{H}_G, \ell_\infty\right) - n\left(\frac{\epsilon}{24} - \frac{\lambda_n\|f_{G^*}^*\|^2}{12}\right)^2\right\} \tag{25}$$

By using Lemma 3, we can bound the covering number for $\mathcal{H}_G$ from above and obtain the following inequality.

$$P\left(|\widehat{\mathcal{R}}_g(\hat{f}_G) - \mathcal{R}_g(f_{G^*}^*)| > \epsilon\right) \leq 12n \cdot \exp\left\{\sum_{\boldsymbol{u} \in G} \ln \mathcal{N}\left(\frac{\epsilon}{12|G|}, \mathcal{H}_{\boldsymbol{u}}, \ell_\infty\right) - \frac{\epsilon^2 n}{144}\right\} +$$

$$12n \cdot \exp\left\{\sum_{\boldsymbol{u} \in G} \ln \mathcal{N}\left(\frac{\epsilon}{12|G|}, \mathcal{H}_{\boldsymbol{u}}, \ell_\infty\right) - n\left(\frac{\epsilon}{24} - \frac{\lambda_n\|f_{G^*}^*\|^2}{12}\right)^2\right\}.$$

$\square$

**Theorem 2.** *Let $\mathcal{G}^a$ be the set of all amiable group structures. For any $\epsilon > 0$ and $n > 2/\epsilon^2$, we have*

$$P\left(\sup_{G \in \mathcal{G}^a} |\widehat{\mathcal{R}}_g(\hat{f}_G^\lambda) - \mathcal{R}_g(f_{G^*}^*)| > \epsilon\right) \leq 12n|\mathcal{G}^a| \cdot \left[\exp\left\{\max_{G \in \mathcal{G}^a} \ln \mathcal{N}\left(\frac{\epsilon}{12}, \mathcal{H}_G, d_\infty\right) - \frac{\epsilon^2 n}{144}\right\}\right.$$

$$\left. + \exp\left\{\max_{G \in \mathcal{G}^a} \ln \mathcal{N}\left(\frac{\epsilon}{12}, \mathcal{H}_G, d_\infty\right) - n\left(\frac{\epsilon}{24} - \frac{\lambda_n\|f_{G^*}^*\|^2}{12}\right)^2\right\}\right] \tag{26}$$

*Proof.* Denote $\mathcal{D}_{G,\epsilon}^{(n)} = \left\{(\boldsymbol{x}_i, y_i)_{i=1}^n \in \mathcal{X} \times \mathcal{Y} \,\middle|\, |\widehat{\mathcal{R}}(\hat{f}_G, g) - \mathcal{R}(f_{G^*}^*, g)| > \epsilon\right\}$, then we have

$$P\left(\bigcup_{G \in \mathcal{G}^a} \mathcal{D}_{G,\epsilon}\right) \leq \sum_{G \in \mathcal{G}^a} P\left(\mathcal{D}_{G,\epsilon}\right)$$

$$\leq |\mathcal{G}^a| 12n \exp\left\{\max_{G \in \mathcal{G}^a} \ln \mathcal{N}\left(\frac{\epsilon}{12}, \mathcal{H}_G, \ell_\infty\right) - \frac{\epsilon^2 n}{144}\right\} +$$

$$|\mathcal{G}^a| 12n \exp\left\{\max_{G \in \mathcal{G}^a} \ln \mathcal{N}\left(\frac{\epsilon}{12}, \mathcal{H}_G, \ell_\infty\right) - n\left(\frac{\epsilon}{24} - \frac{\lambda\|f_{G^*}^*\|^2}{12}\right)^2\right\}$$

where the second inequality is due to the proof of Proposition 4. $\square$

**Theorem 3.** *(i) For a non-amiable group structure $G' \in \mathcal{G} \setminus \mathcal{G}^a$, there exists a constant $C > 0$ such that $|\widehat{\mathcal{R}}_g(\hat{f}_{G'}^\lambda) - \mathcal{R}_g(f_{G^*}^*)|$ converges to $C$ in probability. (ii) There exits a constant $\tilde{C}$ such that $P(|\widehat{\mathcal{R}}_g(\hat{f}_{G'}^\lambda) - \mathcal{R}_g(f_{G^*}^*)| > \tilde{C}$ for all $G' \in \mathcal{G} \setminus \mathcal{G}^a)$ goes to 1 as $n$ goes to infinity.*

*Proof.* We start with the following triangle inequality

$$|\widehat{\mathcal{R}}_g(\hat{f}_{G'}) - \mathcal{R}_g(f^*_{G^*})| \leq |\widehat{\mathcal{R}}_g(\hat{f}_{G'}) - \mathcal{R}_g(\hat{f}_{G'})| + |\mathcal{R}_g(\hat{f}_{G'}) - \mathcal{R}_g(f^*_{G^*})|. \qquad (27)$$

The first term on the RHS can be bounded by using the same uniform convergence bound (12.135) in [20]. For any $\epsilon > 0$ and all $n > 2/\epsilon^2$,

$$\mathbb{P}\left(|\widehat{\mathcal{R}}_g(\hat{f}_{G'}) - \mathcal{R}_g(\hat{f}_{G'})| > \epsilon\right) \leq 12n \cdot \mathbb{E}\left[\mathcal{N}\left(\frac{\epsilon}{6}, \mathcal{H}_{G'}, \ell_\infty^{X'}\right)\right] \exp\left\{-\frac{\epsilon^2 n}{36}\right\}$$

$$\leq 12n \cdot \exp\left\{\ln\mathcal{N}\left(\frac{\epsilon}{6}, \mathcal{H}_{G'}, \ell_\infty\right) - \frac{\epsilon^2 n}{36}\right\}. \qquad (28)$$

In order to derive an upper bound for the second term, we first decompose each risk into bias and variance. According to [6], the risk of the empirical estimate of $\hat{f}_{G'}$ can be decomposed as

$$\mathcal{R}_g(f_{G'}) = \int_{\mathcal{X} \times \mathcal{Y}} \left(g(y) - \hat{f}_{G'}(x)\right)^2 dP_{XY}$$

$$= \int_{\mathcal{X} \times \mathcal{Y}} \left(g(y) - f_{Y|X}(x)\right)^2 dP_{XY} + \int_{\mathcal{X} \times \mathcal{Y}} \left(f_{X|Y}(x) - \hat{f}_{G'}(x)\right)^2 dP_{XY}, \qquad (29)$$

where $f_{X|Y}(x) := \int_{\mathcal{Y}} g(y) dP_{Y|X}$ is the optimal regression function.

By assuming $f_{X|Y}(x) = f^*_{G^*}$ (this is the assumption we use throughout this chapter), we have

$$|\mathcal{R}_g(\hat{f}_{G'}) - \mathcal{R}_g(f^*_{G^*})| = \int_{\mathcal{X} \times \mathcal{Y}} \left(f^*_{G^*}(x) - \hat{f}_{G'}(x)\right)^2 dP_{XY} \qquad (30)$$

According to Theorem 2.1 in [13], we have the following decompositions for the two function on the RHS of (30):

$$f^*_{G^*} = \sum_{\boldsymbol{u} \subseteq \{1,\ldots,p\}} f^*_{G^*,\boldsymbol{u}} \quad \text{with} \quad f^*_{G^*,\boldsymbol{u}} := \sum_{\boldsymbol{v} \subseteq \boldsymbol{u}} (-1)^{|\boldsymbol{u}| - |\boldsymbol{v}|} \, \mathrm{P}_{\{1,\ldots,p\}\setminus\boldsymbol{v}}(f^*_{G^*}),$$

$$\hat{f}_{G'} = \sum_{\boldsymbol{u} \subseteq \{1,\ldots,p\}} \widehat{f}_{G',\boldsymbol{u}} \quad \text{with} \quad \widehat{f}_{G',\boldsymbol{u}} := \sum_{\boldsymbol{v} \subseteq \boldsymbol{u}} (-1)^{|\boldsymbol{u}| - |\boldsymbol{v}|} \, \mathrm{P}_{\{1,\ldots,p\}\setminus\boldsymbol{v}}(\widehat{f}_{G'}).$$

Since $G'$ is an non-amiable group structure, there is at least one subset[1] of $\boldsymbol{u} \subseteq \{1,\ldots,p\}$ such that $f^*_{G^*,\boldsymbol{u}} \neq \hat{f}_{G',\boldsymbol{u}}$. Let $C = \min_{\boldsymbol{u} \subseteq \{1,\ldots,p\}} \int_{\mathcal{X} \times \mathcal{Y}} \left(f^*_{G^*,\boldsymbol{u}} - \hat{f}_{G',\boldsymbol{u}}\right)^2 dP_{XY} > 0$ and denote $\boldsymbol{u}^c = \{1,\ldots,p\} \setminus \boldsymbol{u}$, then we have

$$\int_{\mathcal{X} \times \mathcal{Y}} \left(f^*_{G^*}(\boldsymbol{x}) - \hat{f}_{G'}(\boldsymbol{x})\right)^2 dP_{XY}$$

$$= \int_{\mathcal{X}_{\boldsymbol{u}} \times \mathcal{Y}} \left(f^*_{G^*,\boldsymbol{u}}(\boldsymbol{x}_{\boldsymbol{u}}) - \hat{f}_{G',\boldsymbol{u}}(\boldsymbol{x}_{\boldsymbol{u}})\right)^2 dP_{X_{\boldsymbol{u}}Y} + \int_{\mathcal{X}_{\boldsymbol{u}^c} \times \mathcal{Y}} \left(f^*_{G^*,\boldsymbol{u}^c}(\boldsymbol{x}_{\boldsymbol{u}^c}) - \hat{f}_{G',\boldsymbol{u}^c}(\boldsymbol{x}_{\boldsymbol{u}^c})\right)^2 dP_{X_{\boldsymbol{u}^c}Y}$$

$$\geq C + \int_{\mathcal{X} \times \mathcal{Y}} \left(f^*_{G^*,\boldsymbol{u}^c}(\boldsymbol{x}_{\boldsymbol{u}^c}) - \hat{f}_{G',\boldsymbol{u}^c}(\boldsymbol{x}_{\boldsymbol{u}^c})\right)^2 dP_{XY}$$

$$> 0. \qquad (31)$$

where the first equality is due to the orthogonality possessed by a direct sum Hilbert space.

By using (27), (28), (30) and (31), we can obtain

$$P\left(|\widehat{\mathcal{R}}_g(\hat{f}_{G'}) - \mathcal{R}_g(f^*_{G^*})| > \epsilon + C\right) \leq 12n \cdot \exp\left\{\ln\mathcal{N}\left(\frac{\epsilon}{6}, \mathcal{H}_{G'}, \ell_\infty\right) - \frac{\epsilon^2 n}{36}\right\} \qquad (32)$$

$\square$

**Theorem 4.** *Let $\lambda_n * n \to 0$. By choosing a proper tuning parameter $\mu > 0$ for the structural penalty, the estimated group structure $\hat{G}$ is consistent for the intrinsic group additive structure $G^*$, that is, $P(\hat{G} = G^*)$ goes to one as the sample size $n$ goes to infinity.*

*Proof.* According to Theorem 3, by choosing $\epsilon < C$, an agreeable group structure will be chosen with high probability.

For an amiable group structure, let $\epsilon_1 = |\hat{\mathcal{R}}_g(\hat{f}_G^\lambda) - \mathcal{R}_g(f_{G^*}^*)|$ and $\epsilon_2 = \mu\mathcal{C}(G) - \mu\mathcal{C}(G^*)$. Since $\mathcal{C}(G) > \mathcal{C}(G^*)$ when $G$ is not the true group structure, we have $\epsilon_2 > 0$. Because $\epsilon_1$ converges to 0 in probability. Thus the true group structure $G^*$ will be picked with high probability if Problem (7) is solved. $\qquad\square$

**Lemma 1.** *Let $S, T : \mathcal{F}_1 \to \mathcal{F}_2$ be operators in Banach spaces and $\epsilon_1, \epsilon_2 > 0$. Then we have*
$$\mathcal{N}\left(\epsilon_1 + \epsilon_2, T + S\right) \leq \mathcal{N}\left(\epsilon_1, S\right) \cdot \mathcal{N}\left(\epsilon_2, T\right).$$

**Lemma 2.** *For all $\epsilon > 0$ and all $n > 2/\epsilon^2$,*
$$P\left(|\mathcal{R}_g\left(\hat{f}_G\right) - \mathcal{R}_g\left(f_{G^*}^*\right)| > \frac{\epsilon}{2}\right) \leq 12n \cdot \ln\mathbb{E}\left[\mathcal{N}\left(\frac{\epsilon^2}{12}, \mathcal{H}_G, \ell_\infty^{X'}\right)\right] \exp\left\{-n\left(\frac{\epsilon}{24} - \frac{\lambda_n\|f_{G^*}^*\|^2}{12}\right)^2\right\}.$$

*Proof.* Due to the uniform convergence bound (12.135) in [20], given $\hat{f}_{G,}$, we have for all $\epsilon > 0$ and all $n \geq 2/\epsilon^2$,
$$P\left(|\hat{\mathcal{R}}_g\left(\hat{f}_G\right) - \mathcal{R}_g\left(\hat{f}_G\right)| > \epsilon\right) \leq 12n \cdot \mathbb{E}\left[\mathcal{N}\left(\frac{\epsilon^2}{12}, \mathcal{H}_G, \ell_\infty^{X'}\right)\right] \exp\left\{-\frac{\epsilon^2 n}{36}\right\}.$$

By setting $\delta = 12n \cdot \mathbb{E}\left[\mathcal{N}\left(\frac{\epsilon^2}{12}, \mathcal{H}_G, \ell_\infty^{X'}\right)\right] \exp\left\{-\frac{\epsilon^2 n}{36}\right\}$ and solve for $\epsilon$, we have
$$\epsilon = 6n^{-1/2}\left(\ln 12n + \ln\mathbb{E}\left[\mathcal{N}\left(\frac{\epsilon^2}{12}, \mathcal{H}_G, \ell_\infty^{X'}\right)\right] - \ln\delta\right)^{1/2}.$$

Equivalently with probability at least $1 - \delta$,
$$|\hat{\mathcal{R}}_g\left(\hat{f}_G\right) - \mathcal{R}_g\left(\hat{f}_G\right)| \leq 6n^{-1/2}\left(\ln 12n + \ln\mathbb{E}\left[\mathcal{N}\left(\frac{\epsilon^2}{12}, \mathcal{H}_G, \ell_\infty^{X'}\right)\right] - \ln\delta\right)^{1/2}.$$

Due to the symmetry of the above bound, we have with probability at least $1 - \delta$,
$$\mathcal{R}_g\left(\hat{f}_G\right) \leq \hat{\mathcal{R}}_g\left(\hat{f}_G\right) + 6n^{-1/2}\left(\ln 12n + \ln\mathbb{E}\left[\mathcal{N}\left(\frac{\epsilon^2}{12}, \mathcal{H}_G, \ell_\infty^{X'}\right)\right] - \ln\delta\right)^{1/2}$$
$$\leq \hat{\mathcal{R}}_g\left(\hat{f}_G\right) + \lambda\|\hat{f}_G\|^2 + 6n^{-1/2}\left(\ln 12n + \ln\mathbb{E}\left[\mathcal{N}\left(\frac{\epsilon^2}{12}, \mathcal{H}_G, \ell_\infty^{X'}\right)\right] - \ln\delta\right)^{1/2}$$
$$\leq \hat{\mathcal{R}}_g\left(f_{G^*}^*\right) + \lambda\|f_{G^*}^*\|^2 + 6n^{-1/2}\left(\ln 12n + \ln\mathbb{E}\left[\mathcal{N}\left(\frac{\epsilon^2}{12}, \mathcal{H}_G, \ell_\infty^{X'}\right)\right] - \ln\delta\right)^{1/2}$$
$$\leq \mathcal{R}_g\left(f_{G^*}^*\right) + \lambda\|f_{G^*}^*\|^2 + 12n^{-1/2}\left(\ln 12n + \ln\mathbb{E}\left[\mathcal{N}\left(\frac{\epsilon^2}{12}, \mathcal{H}_G, \ell_\infty^{X'}\right)\right] - \ln\delta\right)^{1/2}$$

where the third inequality is due to the definition of $\hat{f}_G$ as the minimizer of the empirical problem. We applied the uniform convergence bound twice, one for the first inequality and the other for the last inequality.

Since it is always true that $\mathcal{R}_g\left(f_{G^*}^*\right) \leq \mathcal{R}_g\left(\hat{f}_G\right)$, we have the symmetric upper bound with probability $1 - \delta$,
$$|\mathcal{R}_g\left(\hat{f}_G\right) - \mathcal{R}_g\left(f_{G^*}^*\right)| \leq$$
$$\lambda\|f_{G^*}^*\|^2 + 12n^{-1/2}\left(\ln 12n + \ln\mathbb{E}\left[\mathcal{N}\left(\frac{\epsilon^2}{12}, \mathcal{H}_G, \ell_\infty^{X'}\right)\right] - \ln\delta\right)^{1/2}.$$

By setting $\lambda\|f_{G^*}^*\|^2 + 12n^{-1/2}\left(\ln 12n + \ln \mathbb{E}\left[\mathcal{N}\left(\frac{\epsilon^2}{12}, \mathcal{H}_G, \ell_\infty^{X'}\right)\right] - \ln\delta\right)^{1/2} = \epsilon/2$ and solve for $\delta$, we have

$$\delta = 12n \cdot \ln \mathbb{E}\left[\mathcal{N}\left(\frac{\epsilon^2}{12}, \mathcal{H}_G, \ell_\infty^{X'}\right)\right] \exp\left\{-n\left(\frac{\epsilon}{24} - \frac{\lambda_n\|f_{G^*}^*\|^2}{12}\right)^2\right\}$$

Thus the bound for the second term is for all $\epsilon > 0$ and all $n > 2/\epsilon^2$,

$$P\left(|\mathcal{R}_g\left(\hat{f}_G\right) - \mathcal{R}_g\left(f_{G^*}^*\right)| > \frac{\epsilon}{2}\right) \leq$$

$$12n \cdot \ln \mathbb{E}\left[\mathcal{N}\left(\frac{\epsilon^2}{12}, \mathcal{H}_G, \ell_\infty^{X'}\right)\right] \exp\left\{-n\left(\frac{\epsilon}{24} - \frac{\lambda_n\|f_{G^*}^*\|^2}{12}\right)^2\right\}$$

$\square$

The following Lemma is taken from Lemma 1 in [12], which shows the relationship between the covering number of the direct sum of two operators and the covering numbers of the individual operators.

**Lemma 3.** *Let $S, T : \mathcal{B}_1 \to \mathcal{B}_2$ be operators in real Banach spaces and $\epsilon, \delta > 0$. Then,*

$$\mathcal{N}\left(\epsilon + \delta, T + S\right) \leq \mathcal{N}\left(\epsilon, T\right) \cdot \mathcal{N}\left(\delta, S\right).$$

## B. Complete Simulation Results

| $\mu$ | $\alpha$ | M1 | M2 | M3 | M4 | M5 |
|---|---|---|---|---|---|---|
| 1.0000e-10 | 1.00 | 0 | 0 | 0 | 0 | 100 |
| 1.0000e-10 | 2.00 | 0 | 0 | 0 | 0 | 100 |
| 1.0000e-10 | 3.00 | 0 | 0 | 0 | 0 | 100 |
| 1.0000e-10 | 4.00 | 0 | 0 | 0 | 0 | 99 |
| 1.0000e-10 | 5.00 | 0 | 0 | 0 | 0 | 10 |
| 1.0000e-10 | 6.00 | 0 | 0 | 0 | 0 | 0 |
| 1.0000e-10 | 7.00 | 0 | 0 | 0 | 0 | 0 |
| 1.0000e-10 | 8.00 | 0 | 0 | 0 | 0 | 0 |
| 1.0000e-10 | 9.00 | 0 | 0 | 0 | 0 | 0 |
| 1.0000e-10 | 10.00 | 0 | 0 | 0 | 0 | 0 |
| 1.1180e-08 | 1.00 | 0 | 0 | 0 | 0 | 100 |
| 1.1180e-08 | 2.00 | 0 | 0 | 0 | 0 | 98 |
| 1.1180e-08 | 3.00 | 0 | 0 | 0 | 0 | 0 |
| 1.1180e-08 | 4.00 | 0 | 0 | 0 | 0 | 0 |
| 1.1180e-08 | 5.00 | 0 | 0 | 0 | 0 | 0 |
| 1.1180e-08 | 6.00 | 0 | 0 | 0 | 0 | 0 |
| 1.1180e-08 | 7.00 | 0 | 0 | 0 | 0 | 0 |
| 1.1180e-08 | 8.00 | 0 | 0 | 0 | 1 | 0 |
| 1.1180e-08 | 9.00 | 0 | 0 | 0 | 77 | 0 |
| 1.1180e-08 | 10.00 | 0 | 0 | 0 | 92 | 0 |
| 1.2500e-06 | 1.00 | 0 | 0 | 0 | 0 | 100 |
| 1.2500e-06 | 2.00 | 0 | 0 | 0 | 0 | 0 |
| 1.2500e-06 | 3.00 | 14 | 0 | 0 | 84 | 0 |
| 1.2500e-06 | 4.00 | 81 | 3 | 4 | 99 | 0 |
| 1.2500e-06 | 5.00 | 90 | 77 | 77 | 99 | 0 |
| 1.2500e-06 | 6.00 | 94 | 92 | 90 | 99 | 0 |
| 1.2500e-06 | 7.00 | 96 | 96 | 95 | 100 | 0 |
| 1.2500e-06 | 8.00 | 98 | 97 | 96 | 100 | 0 |
| 1.2500e-06 | 9.00 | 98 | 97 | 97 | 100 | 0 |
| 1.2500e-06 | 10.00 | 100 | 97 | 97 | 100 | 0 |
| 1.3975e-04 | 1.00 | 0 | 0 | 0 | 0 | 100 |
| 1.3975e-04 | 2.00 | 0 | 95 | 93 | 100 | 0 |
| 1.3975e-04 | 3.00 | 100 | 95 | 92 | 90 | 0 |
| 1.3975e-04 | 4.00 | 100 | 28 | 23 | 9 | 0 |
| 1.3975e-04 | 5.00 | 100 | 13 | 12 | 3 | 0 |
| 1.3975e-04 | 6.00 | 100 | 5 | 7 | 3 | 0 |
| 1.3975e-04 | 7.00 | 100 | 0 | 0 | 2 | 0 |
| 1.3975e-04 | 8.00 | 100 | 0 | 0 | 0 | 0 |
| 1.3975e-04 | 9.00 | 100 | 0 | 0 | 0 | 0 |
| 1.3975e-04 | 10.00 | 100 | 0 | 0 | 0 | 0 |
| 1.5625e-02 | 1.00 | 0 | 0 | 0 | 0 | 100 |
| 1.5625e-02 | 2.00 | 0 | 0 | 0 | 100 | 0 |
| 1.5625e-02 | 3.00 | 100 | 0 | 0 | 0 | 0 |
| 1.5625e-02 | 4.00 | 100 | 0 | 0 | 0 | 0 |
| 1.5625e-02 | 5.00 | 100 | 0 | 0 | 0 | 0 |
| 1.5625e-02 | 6.00 | 100 | 0 | 0 | 0 | 0 |
| 1.5625e-02 | 7.00 | 100 | 0 | 0 | 0 | 0 |
| 1.5625e-02 | 8.00 | 100 | 0 | 0 | 0 | 0 |
| 1.5625e-02 | 9.00 | 100 | 0 | 0 | 0 | 0 |
| 1.5625e-02 | 10.00 | 100 | 0 | 0 | 0 | 0 |

Table 3: Frequencies that the true group structures are selected under different parameter pairs for the five models. Exhaustive search algorithm without parameter turning.

Figure 3: The 3D surface of the frequencies (out of 100) that the true group structures are identified for the five chosen models in Table 1 over the entire parameters grid. Given a $(\mu, \alpha)$ pair, the penalized goodness of fit is calculated for all group structures. We recorded each time the true group structure is identified. The values of $\mu$ are reported in log-scale. Each surface plot is accompanied with three contour plots as the 2D projections of the surface to enhance the effect of the visualization.

Figure 4: The 3D surfaces of the frequencies (out of 100) that the true group structures are identified for the five chosen models in Table 1 over the entire parameter grids. The training procedure uses a separate validation data set to select the optimal tuning parameters $(\mu, \alpha)$. The values of $\mu$ are reported in log-scale. Each surface plot is accompanied with three contour plots as the 2D projections of the surface to enhance the effect of the visualization.

Figure 5: The 3D surfaces of the frequencies (out of 100) that the true group structures are identified for the five chosen models in Table 1 over the entire parameter grids. The training uses the backward stepwise algorithm and the procedure uses a separate validation data set to select the optimal tuning parameters $(\mu, \alpha)$. The values of $\mu$ are reported in log-scale. Each surface plot is accompanied with three contour plots as the 2D projections of the surface to enhance the effect of the visualization.

## C. Complete Results of Real Data Applications

**Application 1: Boston Housing Data**

Figure 6: The results of applying the backward step-wise algorithm on Boston Housing data with 10-fold CV. The 3D surfaces shows the average validation error over the entire grid of $(\mu, \alpha)$ pairs. The surface plot is accompanied with three contour plots as the 2D projections of the surface to enhance the effect of the visualization.

**Application 2: Communities and Crime Data**

Figure 7: Selected results for the communities and crime data where the number of murders is the response. The blue dots are the transformed observation of the predictor variable. The red line is the estimated function.