[Reviews · NeurIPS 2017]

Reviewer 1



The paper considers kernel regression in the high dimensional setting. The starting point of the paper is the "additive model", which consider regressors on the form f = sum_p f_p(X_p) i.e. correlations between different dimensions are ignored. Obviously it is problematic to ignore such correlations so the authors instead consider the "group additive model", where different dimensions are grouped, such that correlations between some dimensions can be modeled. The authors provide a formal treatment of this setting, and provide some (somewhat) ad hoc algorithms for finding optimal groupings. Results are presented for small synthetic and real-world cases. For me, the key issue with the paper is that it does a rather poor job of describing related work, so it is difficult for me (as a non-expert) to determine the true novelty of the work. The task of finding optimal groups in data is e.g. also considered in "structured sparsity" and when learning graphical models/Bayesian networks. Yet, the authors make no mention of these fields. Likewise, the problem can be seen as an instance of "feature selection", but again this is not mentioned. Effectively, the related work described in the paper boils down to a single KDD-2015 paper [8], which is not enough for me to determine the actual novelty of paper. That being said, the theoretical treatment does strike me as novel. The authors end up defining a simple complexity measure on a grouping (this measure favors many small groups). This is derived as an upper bound on the covering number. This seems sensible enough. The downside, is that this measure does not lend itself to easy optimization, so the authors either propose an exhaustive search (which scales poorly) or a greedy method (akin to forward feature selection). While I acknowledge that discrete functions (such as one describing optimal groupings) are generally difficult to optimize, I must say I was somewhat disappointed that the authors hadn't arrived at a measure that could be optimized more efficiently than just using cross-validation. The authors mention that they have been "investigating and comparing different measures" (line 174), but they provide no further insights, so it is hard for me to determine the suitability of the chosen measure. Experimentally, the authors provide simple synthetic validation of their framework along with a simple example on the Boston Housing data. They provide no baseline and do not compare with other methods. Again, it is difficult for me to determine if the presented results are good as I have nothing to compare with. == Post rebuttal == The authors clarified some of my concerns in their rebuttal. I still think the experiments are thin, and that the authors did an unacceptably poor job of relating their work to ideas in machine learning. However, I do acknowledge that the paper has valid contributions and have improved my score somewhat. I strongly encourage the authors to (at least) improve their discussion of related work when the paper is published.

Reviewer 2



Note: Since the supplement appears to include the main paper, I simply reviewed that, and all line numbers below correspond to the supplement. Summary: This studies group-additive nonparametric regression models, in which, for some partitioning of the predictor variables, the regression function is additive between groups of variables; this model interpolates between the fully nonparametric model, which is difficult to fit, and the additive model, which is sometimes too restrictive. Specifically, the paper studies the problem where the group structure is not known in advance and must be learned from the data. To do this, the paper proposes a novel penalty function, based on the covering numbers of RKHS balls, which is then added to the kernel ridge regression objective. This results in an objective that can be optimized over both the group structure (which, together with the kernel determines a function space via direct sum of RKHSs over each group of variables) and the regression estimate within each group. Two algorithms are presented for approximately solving this compound optimization problem, and then theoretical results are presented showing (a) the rate at which the empirical risk of the estimate approaches the true risk of the true optimum, and (b) consistency of the group structure estimate, in that the probability it matches the true group structure approaches 1 as n -> infinity. Finally, experimental results are presented on both synthetic and real data. Main Comments: The key innovation of the paper appears to be recognizing that the complexity of a particular group structure can be quantified in terms of covering numbers of the direct sum space. The paper is comprehensive, including a well-motivated and novel method, and reasonably solid theoretical and empirical results. I'm not too familiar with other approaches to fitting models between the additive and nonparametric models, but, assuming the discussion in Lines 31-50 is fairly, complete, this paper seems like a potentially significant advance. As noted in the Discussion section, the main issue with the method appears to be difficulty solving the optimization problem over group structure when the number of variables in large. The paper is also fairly clearly written, aside from a lot of typos. Minor Comments/Questions: Just curious: is there any simple characterization of "interaction" between two variables that doesn't rely on writing the whole model in Equation (1)? Line 169: I don't quite understand the use of almost surely here. Is this meant as n -> infinity? Equation (3) is missing a summation (over i). Line 205: Perhaps "translation invariant kernel" is a more common term for this than "convolutional kernel" Typos: Line 145: "on RHS" should be "on the RHS" Line 152: "not only can the bias of \hat f_{\lambda,G} reduces" should be "not only can the bias of \hat f_{\lambda,G} reduce" Line 160: "G^* exists and unique" should be "G^* exists and is unique" Line 247: "turning parameters" should be "tuning parameters" Algorithm 2: Line 1: "State with" should be "start with" Line 251: "by compare" should be "by comparing".